# The impact of human behavioral adaptation stratified by immune status on COVID-19 spread with application to South Korea

**Sileshi Sintayehu Sharbayta**[1]*, **Youngji Jo**[1]*, **Jaehun Jung**[2], **Bruno Buonomo**[3]

**1** Department of Public Health Sciences, School of Medicine, University of Connecticut Health Center, Farmington, Connecticut, United States of America, **2** Department of Preventive Medicine, Korea University College of Medicine, Seoul, South Korea, **3** Department of Mathematics and Applications, University of Naples Federico II, Naples, Italy

* jo@uchc.edu (YJ); sharbayta@uchc.edu (SSS)

## Abstract

As the COVID-19 pandemic continues with ongoing variant waves and vaccination efforts, population-level immunity and public risk perceptions have shifted. This study presents a behavioral transmission model to assess how virus spread and care-seeking behavior differ based on individuals' immunity status. We categorized the population into two groups: "partially immune" and "susceptible," which influenced their response to vaccination and testing, as well as their prioritization of information related to disease prevalence and severity. Using COVID-19 data from South Korea (February 1, 2022 – May 31, 2022), we calibrated our model to explore these dynamics. Simulation results suggest that increasing reactivity to information among partially immune individuals to the same level as susceptible individuals could reduce peak active cases by 33%. Conversely, if partially immune individuals shift their risk perception focus from prevalence (90% prevalence vs. 10% severity) to severity (90% severity vs. 10% prevalence), the peak in active cases could increase by 57%. These findings highlight the need for adaptive vaccination and testing strategies as public risk perceptions evolve due to prior exposures and vaccinations. As new variant waves emerge in the post-pandemic endemic era, our study offers insights into how immunity-based behavioral differences can shape future infection peaks.

## Introduction

COVID-19 increasingly appears likely to enter long-term circulation and become endemic, necessitating regular vaccinations with updated vaccines, similar to seasonal influenza [1,2]. Changes in risk perceptions during the pandemic affect behaviors, including testing and willingness to vaccinate [3]. Patterns of COVID-19 transmission shape the subsequent patterns of behavioral responses to the disease and, in turn, are shaped by such responses. Some models have been developed to

**Data availability statement:** This study used official COVID-19 data from South Korea obtained from Our World in Data [32]. The code used to generate the results is available at: https://github.com/UConn-Health-Disease-Modeling/COVID-19-Behavior.

**Funding:** This research was supported by the Infectious Disease Medical Safety Project funded by the Ministry of Health & Welfare, Republic of Korea (Grant No. HG22C0094), and by EU funding under the Next Generation EU–MUR PNRR Extended Partnership on Emerging Infectious Diseases (Project No. PE00000007, INF-ACT) and the PRIN 2020 project (No. 2020JLWP23), "Integrated Mathematical Approaches to Socio–Epidemiological Dynamics." The funders had no role in the study design; data collection, analysis, or interpretation; decision to publish; or preparation of the manuscript.

**Competing interests:** The authors have declared that no competing interests exist.

help policymakers compare interventions such as testing and vaccination [4,5]. Such models, dependent on various uncertain assumptions, attempt to forecast cases, deaths, and medical supplies needs; predict the timing of peaks in cases; and estimate if and when to expect additional waves or surges. Despite the rapid advancements in COVID-19 models and forecast tools, very few models [6–8] directly incorporate adaptive behavioral components to account for changes in risk perceptions, protective behaviors, and compliance with interventions over time, which ultimately influences transmission.

Compliance with testing and willingness to vaccinate significantly affect disease transmission dynamics and can influence policy recommendations. Hence, modeling how voluntary testing compliance and vaccination willingness are influenced by individual immune status/history and available public information (e.g., rumors on vaccine efficacy or level of prevalence/severity) is crucial. Coupling transmission models of infectious diseases with models of behavior adaptation have been a growing field of research [9–11]. More specifically, an information index approach was recently employed in epidemic models to account for the social behavior change due to available information related to the disease status in the population [12–16]. This strategy takes into consideration how the information is distributed; the dependence of human behavior not only on the current knowledge but also on the past state of the disease in the population; and how long it takes for the information to reach the general public (information delay). For example, vaccination behavior changes due to available information about the prevalence has been considered in a meningitis model [14]. In [16], the authors considered a general SISV (Susceptible-Infectious-Susceptible-Vaccinated) model where transmission (social distancing compliance behavior) and vaccination rate (vaccination decision) depend on prevalence and vaccination roll-out, respectively.

In this study, we develop a compartmental model to represent transmission dynamics of COVID-19 structured by two different susceptible populations (naive vs. partially immune due to previous infection or vaccination). We then incorporate the information index approach, where the individuals' compliance of vaccination and testing is based on two different kinds of information, namely information about the level of prevalence and severity of the disease. We also take into account the change of risk perception by differing the weighting on information between prevalence and severity among people who are partially immune and not immune (i.e., susceptible) [17,18]. Although our study adopts the information index approach—similar to the methodology used in several previous works—our modeling framework introduces a more nuanced and realistic representation of individual behavioral responses. Most prior studies employing this approach tend to assume a homogeneous population in which all individuals respond uniformly to a single type of information, typically related to disease prevalence. Such models often overlook other critical drivers of behavior, including an individual's immune status or the perceived severity of the disease (e.g., hospitalization and mortality rates) that may lead to different care seeking practice. In contrast, our model differentiates between various sources of information that individuals may consider when deciding whether to voluntarily comply with interventions

such as vaccination or testing. Specifically, we incorporate both prevalence- and severity-related information into the care seeking process. Moreover, we allow for heterogeneous responses based on immune status, acknowledging that immune-naive individuals and those with partial immunity may interpret and act on available information differently. This layered behavioral structure enables a more flexible and realistic simulation of population dynamics during an epidemic, particularly under evolving public perception and varying health risks.

## Materials and methods

### Model formulation

The entire population (N) is divided into seventeen distinct compartments according to individuals' infection and vaccination status. We divide the transmission dynamics into two categories: primary dynamics, which describes disease transmission for susceptible individuals, and secondary dynamics, which describes transmission among partially immune individuals due to vaccination or previous infection. Each dynamic's transmission is discussed below.

*(i) Primary and secondary dynamics*

The primary dynamics consist of seven compartments: susceptible non-immune individuals ($S_1$); primary-series vaccinated individuals ($V_1$); exposed ($E_1$); asymptomatic infectious ($A_1$); symptomatic infectious ($I_1$); tested and detected ($I_{T1}$); and undetected infectious individuals ($I_{U1}$). Susceptible individuals become infected at rate $\lambda_1$ and enter the exposed class. Exposed individuals progress to the asymptomatic or symptomatic compartments at rates $(1-\tau)\epsilon$ and $\tau\epsilon$, respectively, where $\tau$ is the proportion developing symptoms and $\epsilon^{-1}$ is the average latent period.

Individuals in $A_1$ and $I_1$ undergo testing at rates $a_1\xi T_1$ and $T_1$, respectively, moving to the detected class $I_{T1}$, while the remainder enter the undetected class $I_{U1}$. Here, $\xi$ represents the relative testing propensity of asymptomatic individuals, and $T_1$ is the testing rate. Following clinical considerations, symptomatic individuals are assumed to be detected after a one-day delay, while asymptomatic individuals experience a longer average detection delay of $1/a_1$ days ($0 < a_1 < 1$).

The secondary dynamics include eight compartments representing partially immune individuals: $S_2$ and $S_3$ (with vaccination history and infection/vaccination history, respectively); booster-vaccinated individuals ($V_2$); and the corresponding exposed ($E_2$), asymptomatic ($A_2$), symptomatic ($I_2$), tested ($I_{T2}$), and undetected ($I_{U2}$) classes. Immunity levels differ across these groups, leading to reduced infection rates of $(1-\eta_2)\lambda_2$, $(1-\eta_3)\lambda_2$, and $(1-\eta_4)\lambda_2$ for $S_2$, $S_3$, and $V_2$, respectively. The parameters $\eta_2$, $\eta_3$, and $\eta_4$ reflect the protective effects of primary vaccination, prior infection, and booster vaccination after six months, based on recent estimates of immunity durability [19,20]. Individuals vaccinated in the primary series ($V_1$) may be infected at rate $(1-\eta_1)\lambda_2$, where $\eta_1$ denotes the corresponding vaccine effectiveness. Transition from primary to secondary dynamics occurs through vaccination or recovery at rate $\phi$. Individuals in $S_2$ and $S_3$ may receive booster vaccination at rate $F_2$. The progression from $E_2$ to $A_2$ or $I_2$ follows the same structure as in the primary dynamics. Testing processes mirror those in the primary branch, with $A_2$ and $I_2$ tested at rates $a_1\xi T_2$ and $T_2$, respectively, moving to $I_{T2}$, while the remaining individuals enter $I_{U2}$ at rates $a_1(1-\xi T_2)$ and $1-T_2$.

In both the primary and secondary pathways, the $A_{1,2}$ and $I_{1,2}$ compartments represent a brief early infectious stage during which individuals have not yet undergone testing. Individuals in these compartments either transition to the detected or undetected infectious classes. No recovery or mortality occurs during this short phase, and we assume no onward transmission from these stages.

Individuals in the undetected and detected infectious classes ($I_{U1}$, $I_{U2}$, $I_{T1}$, $I_{T2}$) recover at rate $\rho$ and move to the recovered class ($R$). Detected individuals ($I_{T1}$, $I_{T2}$) may require hospitalization at rates $h_1$ and $h_2$, respectively, with $h_2 < h_1$ reflecting reduced severity among partially immune individuals. Hospitalized individuals recover at rate $\rho_h$, while disease-induced mortality occurs in $I_{U1}$, $I_{U2}$, and $H$ at rates $d_1$ and $d_2$. New individuals enter the susceptible class ($S_1$) through birth at rate $\pi$, and natural mortality occurs in all compartments at rate $\mu$. The full model structure is illustrated in Fig 1, and all

 

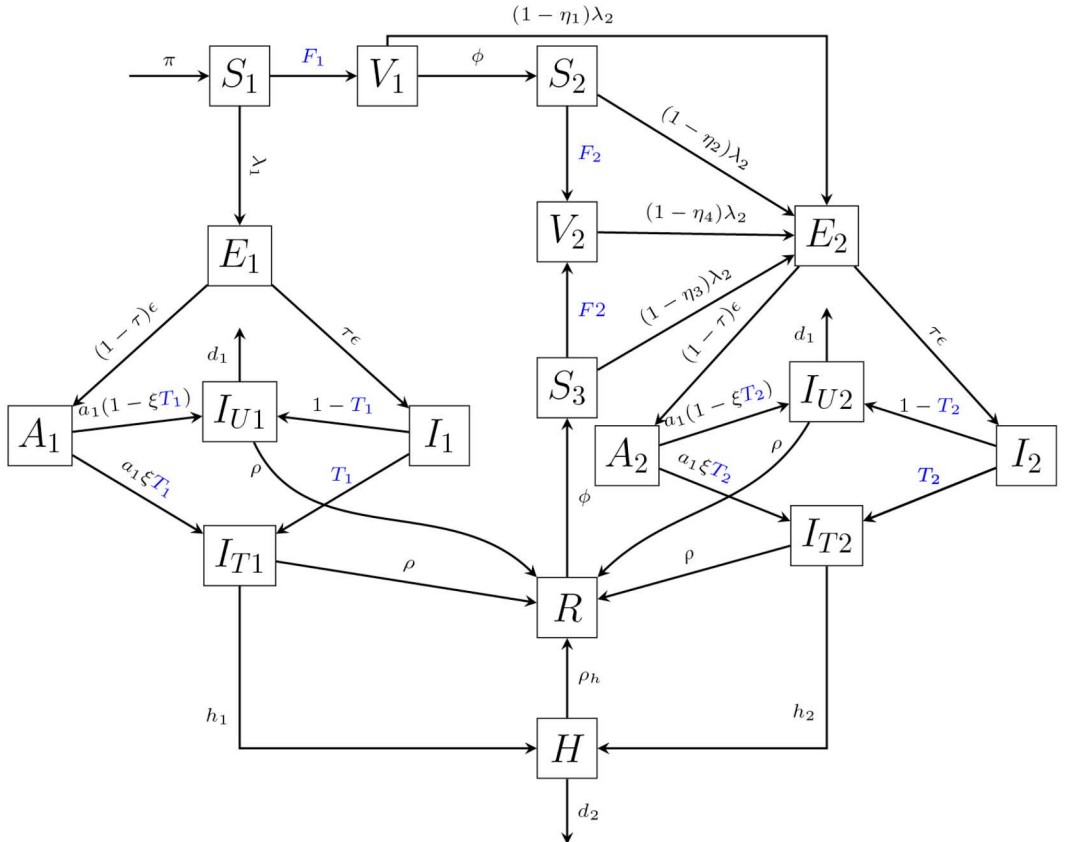

**Fig 1. Transmission dynamics of COVID-19.** The blue-colored parameters are information dependent parameters. $F_1$ is a primary series vaccination rate and $F_2$ is a booster vaccination rate. $T_1$ and $T_2$ are testing rates in primary and secondary dynamics respectively.

compartment and parameter definitions are summarized in Table 1. The formulations of the infection process, information indices, and voluntary vaccination and testing rates are detailed in items (ii)–(iv) below.

*(ii) Force of infection*

Due to immune differences, we assumed the transmission rate in primary and secondary dynamics to be different [21,22]. Given that mandatory quarantine measures are not commonly enforced nowadays, we assume that a certain proportion of people who test positive and all hospital admissions will opt to self-quarantine or isolate themselves, thus they do not contribute to the transmission [23]. Furthermore, due to the brief duration of stay in compartments $A_1$, $A_2$, $I_1$, and $I_2$, the contribution to new infections from individuals in these compartments is assumed to be negligible and is therefore, excluded from the transmission process. With these assumptions, the forces of infection are defined as:

$$\lambda_1 = \beta_1 \frac{I_{U1} + I_{U2} + (1-\delta)(I_{T1} + I_{T2})}{N - (H + \delta(I_{T1} + I_{T2}))}, \quad \lambda_2 = \beta_2 \frac{I_{U1} + I_{U2} + (1-\delta)(I_{T1} + I_{T2})}{N - (H + \delta(I_{T1} + I_{T2}))}, \quad (1)$$

where $N$ is the total population and is given by:

$$N = S_1 + S_2 + S_3 + V_1 + V_2 + E_1 + E_2 + A_1 + A_2 + I_1 + I_2 + H + I_{T1} + I_{T2} + I_{U1} + I_{U2} + R,$$

**Table 1. Symbols and parameters in the model and their description.**

| Variable | Description |
| --- | --- |
| $S_1$ | Non-immune susceptible individuals |
| $S_2$ | Partially-immune (prior vaccinated) susceptible individuals |
| $S_3$ | Partially-immune (prior infected) susceptible individuals |
| $V_1$ | Individuals vaccinated with primary series vaccination |
| $V_2$ | Individuals vaccinated with booster vaccination |
| $E_1$ | Non-immune individuals who are exposed to COVID-19 |
| $A_1$ | Non-immune infected individuals who are asymptomatic |
| $I_1$ | Non-immune infected individuals who are symptomatic |
| $I_{U1}$ | Non-immune infected individuals who are undetected |
| $I_{T1}$ | Non-immune infected individuals who are detected (tested) |
| $E_2$ | Partially-immune individuals who are exposed to COVID-19 |
| $A_2$ | Partially-immune infected individuals who are asymptomatic |
| $I_2$ | Partially-immune infected individuals who are symptomatic |
| $I_{U2}$ | Partially-immune infected individuals who are undetected |
| $I_{T2}$ | Partially-immune infected individuals who are detected (tested) |
| $H$ | Hospitalized individuals |
| $R$ | Recovered individuals |
| Parameter | |
| $\pi$ | Recruitment rate to susceptible class |
| $\mu$ | Natural death rate |
| $\epsilon$ | Progression rate to infectiousness |
| $\tau$ | Proportion of individual that become symptomatic |
| $\rho$ | Recovery rate for undetected & tested classes |
| $\rho_h$ | Recovery rate for hospitalized class |
| $\eta_1$ | Primary series vaccine effectiveness |
| $\eta_2$ | Primary series vaccine effectiveness after 6 months |
| $\eta_3$ | Effectiveness of immunity due to prior infection after 6 months |
| $\eta_4$ | Booster vaccine effectiveness |
| $\phi$ | Progression rate from recovery and primary vaccination classes to secondary dynamics |
| $h_1$ | Hospitalization rate for tested individuals in primary dynamics |
| $h_2$ | Hospitalization rate for tested individuals in secondary dynamics |
| $F_{max}$ | Maximum vaccination rate |
| $F_{10}$ | Mandatory primary series vaccination rate |
| $F_{20}$ | Mandatory booster vaccination rate |
| $T_{max}$ | Maximum testing rate |
| $T_{10}$ | Mandatory testing rate in primary dynamics |
| $T_{20}$ | Mandatory testing rate in secondary dynamics |
| $\frac{1}{a}$ | Average information delay time |
| $D$ | Individuals' reactivity level to prevalence information |
| $B$ | Individuals' reactivity level to severity information |
| $\theta$ | Modification in reactivity to information by immune people relative to non-immune |
| $\delta$ | Percentage of positively tested people who self-quarantine |
| $\alpha_i, i = 1, 2$ | Weight given to the prevalence information relative to severity |

*(Continued)*

**Table 1.** (Continued)

| Variable | Description |
| --- | --- |
| $d_1$ | Disease-induced death rate for undetected individuals |
| $d_2$ | Disease-induced death rate for hospitalized individuals |
| $\beta_1$ | Transmission rate (primary dynamics) |
| $\beta_2$ | Transmission rate (secondary dynamics) |
| $k$ | Information coverage |
| $\xi$ | Testing modification for asymptomatic class relative to symptomatic ones |
| $1/a_1$ | Testing delay for asymptomatic individuals |

and $\beta_1$ and $\beta_2$ are the transmission rates in primary and secondary dynamics respectively, $\delta$ is percentage of tested individuals who quarantine.

*(iii) Information*

Information about the level of disease circulating in a community plays a critical role in shaping individuals' compliance behavior. In the context of our model, such information pertains to either disease prevalence ($\mathcal{V}$) or disease severity ($\mathcal{N}$). The variables $\mathcal{V}$ and $\mathcal{N}$ are the information indices that represent the information or public perception related to disease and are modeled using delayed distributions, as defined below:

$$\mathcal{V}(t) = \int_0^\infty g_1(t-x)ae^{-ax}dx,$$

(2)

and

$$\mathcal{N}(t) = \int_0^\infty g_2(t-x)ae^{-ax}dx,$$

(3)

where $a$ control the rate of memory decay regarding past information, where smaller values of $a$ corresponds to a longer memory, meaning that past information (e.g., historical prevalence or severity data) has a prolonged effect on the public's current perception and behavior while larger values of $a$ indicate a shorter memory, with recent observations having a stronger influence. The functions $g_1(t)$ and $g_2(t)$ represent people's perception of the risk of infection and the level of severity of the disease, respectively (often called message functions). Perceived risk of infection and severity are assumed to depend on the number of detected symptomatic and hospitalized individuals ($I_{T1} + I_{T2} + H$), and number of hospitalized and dead people ($H + d_1(I_{U1} + I_{U2}) + d_2H$), respectively. Thus, the message functions are given by

$$g_1(t) = \frac{k(I_{T1}(t) + I_{T2}(t) + H(t))}{N_0},$$
$$g_2(t) = \frac{k(H(t) + d_1(I_{U1}(t) + I_{U2}(t)) + d_2H(t))}{N_0},$$

(4)

where the quantity $k$ represents information coverage, assumed to be the same for prevalence and severity related information. We define information coverage as the publicly available information about the disease status [14,16] and $N_0$ is the steady state population when there are no disease and disease-induced death (*i.e.*, $N_0 = \frac{\pi}{\mu}$). Undetected people are often unreported and hidden from the public, therefore they are not included in information indices. The formulation in equations (2) and (3) represents that the population's memory about the perceived risk is fading exponentially. The term

$ae^{-at}$ is often known as the *exponential fading kernel* [12]. It represents the weight given to the current and past values of the disease. Utilizing the *linear chain trick* method [24], the integral equations (2) and (3) can be reduced into ordinary differential equations (ODEs), given by

$$\dot{\mathcal{V}} = a\left(k\frac{(I_{T1} + I_{T2} + H)}{N_0} - \mathcal{V}\right),$$
$$\dot{\mathcal{N}} = a\left(k\frac{(H + d_1(I_{U1} + I_{U2}) + d_2 H)}{N_0} - \mathcal{N}\right).$$

(5)

*(iv) Vaccination and testing rates*

The vaccination and testing rates are both described by the sum of two rates: mandatory and voluntary rates. First, the mandatory vaccination and testing rates (modeled as constant) represent the rates for the portion of the population that will be vaccinated or tested regardless of the information. These terms summarize some aspects of vaccine acceptance or test seeking of individuals that are strongly in favor of vaccines or advised to get tested, or specific population groups (e.g., older age groups, teachers or health workers) for which the vaccination or testing is mandatory or strongly recommended by the authorities. These rates are represented by $F_{10}$ (mandatory primary series vaccination rate), $F_{20}$ (mandatory booster vaccination rate) and $T_{10}$ (mandatory testing rate in primary dynamics), $T_{20}$ (mandatory testing rate in secondary dynamics), respectively. These mandatory rates establish a baseline level of vaccination and testing uptake in the population, enabling the model to distinguish this baseline behavior from the additional information-driven voluntary responses. Second, the voluntary rate is a rate for a portion of the population voluntarily choosing to be vaccinated or tested depending on the perceived level of disease prevalence and severity in the society. We use the information index to represent the publicly available information or rumors about the prevalence and severity of the disease. Reported number of people dead and hospitalized is used to represent the level of severity of the disease. In these formulations, we made two assumptions: first, partially immune people have a lower perception of the risk of infection, and second, partially immune and susceptible people prioritize prevalence and severity information differently. The Holling type *II* function, (characterized by saturating, continuous, differentiable, and increasing function), is commonly used to represent the voluntary rate [14]. Based on the above discussion, the vaccination and testing rates are given by:

$$F_1(\mathcal{V}, \mathcal{N}) = F_{10} + (F_{max} - F_{10})\left(\alpha_1\frac{D\mathcal{V}}{1 + D\mathcal{V}} + (1 - \alpha_1)\frac{B\mathcal{N}}{1 + B\mathcal{N}}\right),$$

(6)

$$T_1(\mathcal{V}, \mathcal{N}) = T_{10} + (T_{max} - T_{10})\left(\alpha_1\frac{D\mathcal{V}}{1 + D\mathcal{V}} + (1 - \alpha_1)\frac{B\mathcal{N}}{1 + B\mathcal{N}}\right),$$

(7)

and

$$F_2(\mathcal{V}, \mathcal{N}) = F_{20} + (F_{max} - F_{20})\left(\alpha_2\frac{\theta D\mathcal{V}}{1 + \theta D\mathcal{V}} + (1 - \alpha_2)\frac{\theta B\mathcal{N}}{1 + \theta B\mathcal{N}}\right),$$

(8)

$$T_2(\mathcal{V}, \mathcal{N}) = T_{20} + (T_{max} - T_{20})\left(\alpha_2\frac{\theta D\mathcal{V}}{1 + \theta D\mathcal{V}} + (1 - \alpha_2)\frac{\theta B\mathcal{N}}{1 + \theta B\mathcal{N}}\right),$$

(9)

where $D$ and $B$ are positive factors that adjust the reactivity of individuals to the prevalence and severity, respectively, information, $F_{max}$ and $T_{max}$ represent the maximum vaccination and testing rates, respectively, that can be achieved in the

case of a high level of information coverage about the disease status (or high level of risk perception), $\theta$ represents the reduced reactivity to risk perception by individuals in the second dynamics relative to individuals in primary dynamics. The weight given by individuals to the prevalence information in primary and secondary dynamics is measured by $\alpha_1$ and $\alpha_2$, respectively. The corresponding complementary weights, $1 - \alpha_1$ and $1 - \alpha_2$, are assigned to the severity information.

Based on the above discussions, the system under study is governed by the following non-linear ODEs.

$$
\begin{aligned}
\dot{S}_1 &= \pi - (\lambda_1 + F_1(\mathcal{V}, \mathcal{N}) + \mu)S_1, \\
\dot{S}_2 &= \phi V_1 - ((1 - \eta_2)\lambda_2 + F_2(\mathcal{V}, \mathcal{N}) + \mu)S_2, \\
\dot{S}_3 &= \phi R - ((1 - \eta_3)\lambda_2 + F_2(\mathcal{V}, \mathcal{N}) + \mu)S_3, \\
\dot{V}_1 &= F_1(\mathcal{V}, \mathcal{N})S_1 - \phi V_1 - (1 - \eta_1)\lambda_2 V_1 - \mu V_1, \\
\dot{V}_2 &= F_2(\mathcal{V}, \mathcal{N})S_2 + F_2 S_3 - (1 - \eta_4)\lambda_2 V_2 - \mu V_2, \\
\dot{E}_1 &= \lambda_1 S_1 - (\epsilon + \mu)E_1, \\
\dot{E}_2 &= \lambda_2((1 - \eta_1)V_1 + (1 - \eta_4)V_2 + (1 - \eta_2)S_2 + (1 - \eta_3)S_3) - (\epsilon + \mu)E_2, \\
\dot{A}_j &= (1 - \tau)\epsilon E_j - a_1 A_j, \\
\dot{I}_j &= \tau \epsilon E_j - I_j, \\
\dot{I}_{Tj} &= T_j(\mathcal{V}, \mathcal{N})(I_j + a_1 \xi A_j) - (\rho + \mu + h_j)I_{Tj}, \\
\dot{I}_{Uj} &= (1 - T_j(\mathcal{V}, \mathcal{N}))I_j + a_1(1 - \xi T_j(\mathcal{V}, \mathcal{N}))A_j - (\rho + \mu + d_1)I_{Uj}, \\
\dot{H} &= h_1 I_{T1} + h_2 I_{T2} - \rho_h H - (d_2 + \mu)H \\
\dot{R} &= \rho(I_{U1} + I_{U2} + I_{T1} + I_{T2}) + \rho_h H - (\phi + \mu)R, \\
\dot{\mathcal{V}} &= a\left(\frac{k(I_{T1} + I_{T2} + H)}{N_0} - \mathcal{V}\right) \\
\dot{\mathcal{N}} &= a\left(\frac{k(H + d_1(I_{U1} + I_{U2}) + d_2 H)}{N_0} - \mathcal{N}\right),
\end{aligned}
\tag{10}
$$

with initial conditions:

$$
\begin{aligned}
&S_1(0) > 0, S_2(0) \geq 0, S_3(0) \geq 0, V_j(0) \geq 0, E_j(0) \geq 0, A_j(0) \geq 0, I_j(0) \geq 0, I_{Tj} \geq 0, \\
&I_{Uj} \geq 0, H(0) \geq 0, R(0) \geq 0, \mathcal{V}(0) \geq 0, \mathcal{N}(0) \geq 0,
\end{aligned}
\tag{11}
$$

where $j \in \{1, 2\}$.

## Basic properties

In this section, we investigate the basic characteristics of the model (10), which include ensuring the positivity and boundedness of solutions, computing the disease-free equilibrium, and determining the reproduction number.

*(i) Positivity and boundedness of solutions*

Positivity and boundedness of the solutions can be established in standard ways (see, e.g., [25,26]).

*(ii) Disease-free equilibrium and effective reproduction number*

A disease-free equilibrium point is an equilibrium point at which there are no infected individuals in the population (no disease). Setting all infected compartments of the system (10) to zero and solving the reduced system of equations by equating to zero, we get the disease-free equilibrium point, denoted by $E^0$, and given by

$$E^0 = (S_1^0, S_2^0, S_3^0, V_1^0, V_2^0, I^0),$$
(12)

where

$$S_1^0 = \frac{\pi}{F_{10} + \mu},$$

$$S_2^0 = \frac{\pi\phi F_{10}}{(F_{10} + \mu)(F_{20} + \mu)(\phi + \mu)},$$

$$S_3^0 = 0,$$

$$V_1^0 = \frac{\pi F_{10}}{(F_{10} + \mu)(\phi + \mu)},$$

$$V_2^0 = \frac{\pi\phi F_{10} F_{20}}{\mu(F_{10} + \mu)(F_{20} + \mu)(\phi + \mu)},$$

$$I^0 = (E_1^0, E_2^0, A_1^0, A_2^0, I_1^0, I_2^0, I_{T1}^0, I_{T2}^0, I_{U1}^0, I_{U2}^0, H^0, R^0, D^0, \mathcal{V}^0, \mathcal{N}^0) = \mathbf{0},$$

where $\mathbf{0}$ represents a zero vector of dimension $1 \times 15$.

We used the next-generation matrix method to calculate an effective reproduction number for the model (10). The effective reproduction number is the expected number of new infections caused by an infectious individual in a population where some individuals may no longer be susceptible (due to obtained immunity from prior infection or vaccination) [27]. This method follows the following three steps (for detailed explanations, one can refer to [28,29]):

*Step I:* We sort out the equations for infected compartments ($E_1, E_2, A_1, A_2, I_1, I_2, I_{T1}, I_{T2}, H$) and split the right-hand side of the equations as

$$\mathcal{F}_i - \mathcal{G}_i,$$

where $\mathcal{F}_i$ represents the rate of appearance of new infections in compartment $i$ and $\mathcal{G}_i$ incorporates the remaining terms representing the transition of people into and out of the compartments.

*Step II:* Determine the following matrices that are obtained by linearizing the equations in Step I and evaluating at the disease-free equilibrium.

$$F = \left[\frac{\partial \mathcal{F}_i(E^0)}{\partial x_j}\right] \quad \text{and} \quad G = \left[\frac{\partial \mathcal{G}_i(E^0)}{\partial x_j}\right],$$

where $x$ represents the infected compartments.

The next-generation matrix is defined as

$$FG^{-1}.$$

*Step III:* Find the reproduction number using

$$R_e = \rho(FG^{-1}),$$

where $\rho$ is the spectral radius of the matrix and is defined as the maximum of the absolute values of the eigenvalues of the matrix $FG^{-1}$.

Following the above steps, the effective reproduction number, $R_e$, is given by

$$R_e = \left[ FI_p + FI_s \right],$$

(13)

where

$FI_p$ and $FI_s$ represents the effective reproduction number for primary dynamics and secondary dynamics, respectively, and are given by

$$FI_p = \frac{C_1 \epsilon}{\epsilon + \mu} \left[ \frac{(1-\delta)T_{10}\tau(1-\xi) + T_{10}\xi}{h_1 + \mu + \rho} + \frac{T_{10}\tau(1-\xi) + (1-T_{10}\xi)}{\mu + \rho + d_1} \right],$$

$$FI_s = \frac{C_2 \epsilon}{\epsilon + \mu} \left[ \frac{(1-\delta)T_{20}\tau(1-\xi) + T_{20}\xi}{h_2 + \mu + \rho} + \frac{T_{20}\tau(1-\xi) + (1-T_{20}\xi)}{\mu + \rho + d_1} \right],$$

and,

$$C_1 = \frac{\beta_1 S_1^0}{S_1^0 + S_2^0 + V_1^0 + V_2^0},$$

$$C_2 = \frac{\beta_2}{S_1^0 + S_2^0 + V_1^0 + V_2^0} \left( (1-\eta_2)S_2^0 + (1-\eta_1)V_1^0 + (1-\eta_4)V_2^0 \right),$$

the terms $S_1^0$, $S_2^0$, $V_1^0$, and $V_2^0$ represent the disease-free equilibrium values of the corresponding state variables, as defined in equation (12).

**Remark 1.** *At the disease-free equilibrium we have zero prevalence and severity ($\mathcal{V} = \mathcal{N} = 0$). Therefore, the voluntary contribution evaluates to zero at the disease free equilibrium and the testing/vaccination rates reduce to their baseline (mandatory) values (e.g., $T_1 = T_{10}, T_2 = T_{20}, F_1 = F_{10}, F_2 = F_{20}$). Because the next-generation linearization is computed at the DFE, any terms that multiply $\mathcal{V}$ or $\mathcal{N}$ - i.e., the voluntary components – vanish and do not enter the Jacobian used to form the next-generation matrix. Consequently, the effective reproduction number, $R_e$, depends only on the baseline (constant/ mandatory) rates and not on the voluntary-response parameters. Voluntary parameters nevertheless affect the system dynamics (for example peak size, and transient behavior), as shown in the Result Section (Figs 2b, 5, 6).*

**Theorem 1.** *The disease-free equilibrium, $E^0$, of the model (10) is locally asymptotically stable if $R_e < 1$ and unstable if $R_e > 1$.*

*Proof.* The proof follows from Theorem 2 in [30].  □

*(iii) Stability region for the disease-free equilibrium in $(T_{10}, T_{20})$ plane*

From a vaccination and testing perspective, the parameters that drive the effective reproduction number are those related to the mandatory vaccination and testing rates ($F_{10}, F_{20}, T_{10}, T_{20}$). Thus, by varying these rates, one can achieve the condition $R_e < 1$ (stability of the disease-free equilibrium). Since varying (up to $F_{max}$) the mandatory vaccination rates, $F_{10}$ and $F_{20}$, does not modify $R_e$ beyond one, we vary the mandatory testing rates in primary dynamics ($T_{10}$) and secondary dynamics ($T_{20}$) here to depict their impact on the stability region. The result is shown in Fig 2 panel a, where the shaded region is the region in which $R_e > 1$ (showing instability of the disease-free equilibrium) and the non-shaded region is where $R_e < 1$ (showing stability of the disease-free equilibrium). The testing rate for secondary dynamics ($T_{20}$) is a key driving parameter. Moreover, it is possible to establish threshold values for these parameters, assuming all other model parameters remain fixed. For example, when the mandatory testing rate - $T_{20}$ is less than 0.4 then $R_e > 1$ regardless of the wide variation of the testing rate in primary dynamics. However, if $T_{20} > 0.4$, then the disease spread can be controlled over time ($R_e < 1$). The boundary of the regions ($R_e = 1$) in the figure is where the stability of the disease-free equilibrium changes. Furthermore, Fig 2 panel b illustrates the trajectories of the detected prevalence for parameter values yielding $R_e < 1$ and

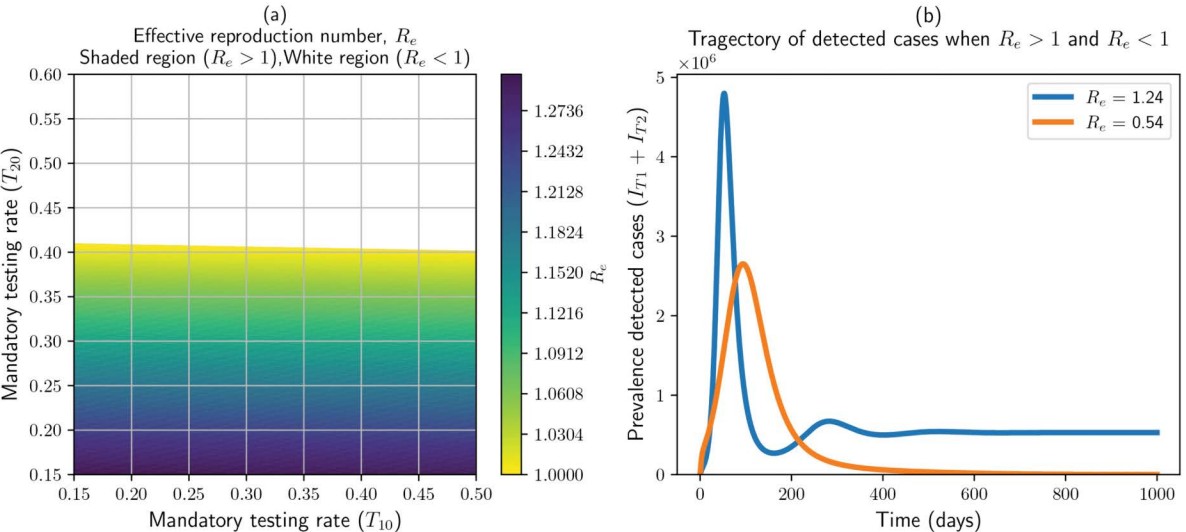

**Fig 2. Panel a: value of the effective reproduction number $R_e$ by varying the mandatory testing rate in primary ($T_{10}$) and secondary ($T_{20}$) dynamics.** The shaded region is a region where $R_e > 1$ and the white region is where $R_e < 1$. Panel b: Trajectories of detected cases for $R_e < 1$ and $R_e > 1$.

$R_e > 1$. As expected, when $R_e < 1$ the prevalence declines to zero, indicating convergence to the disease-free equilibrium, whereas for $R_e > 1$ the prevalence approaches a positive steady state corresponding to an endemic equilibrium. For this demonstration, we selected ($T_{10}, T_{20}$) = (0.5, 0.8) from the white region of the stability plot, which gives $R_e = 0.54$, and ($T_{10}, T_{20}$) = (0.2, 0.2) from the shaded region, which yields $R_e = 1.24$.

## Model fitting and parameter estimation

In this section, we will discuss how we set the model's baseline parameter values. The parameter values in the model are determined in two ways: first, using demographic and epidemiological data obtained from Our World in Data [31] and previous research, which is mentioned as a reference in Table 2, and second, by fitting the model to the Korean COVID-19 vaccination, incidence and mortality data collected during the omicron variant wave with time period from February 01, 2022, to May 31, 2022. We will discuss the details of each method in the following sections.

### (i) Values of known parameters and initial conditions for the model

According to the data from Our World in Data, the estimated population of South Korea in 2022 is $N(0) = 51815808$ (the initial population used in the simulation) and the life expectancy is 83 years [31]. Therefore, the daily natural death rate can be calculated as $\mu = \frac{1}{83 \times 365}$ per person per day and the daily birth rate is obtained by $\pi = \mu \times N(0) = 1710$ individuals per day [37]. The mandatory primary series vaccination rates and the maximum vaccination rate are estimated from the data [31]. We estimated the maximum vaccination rate, $F_{max}$, to be the maximum proportion of daily vaccinated people given by: $F_{max} = \frac{1382042}{51815808} = 0.027$, where the value 1,382,042 is the highest ever recorded vaccination per day for South Korea, obtained from the data in Our World in Data [31]. The mandatory vaccination rate, $F_{10}$, is obtained by calculating the average of the daily proportion of vaccinated people prior to the initial time for our simulation (February 01, 2022), under the assumption that this value can represent the baseline vaccination rate (not influenced by the current level of omicron prevalence or severity) before the omicron wave. Thus, we found $F_{10} = 0.0023$. We assumed a slightly (10%) lower rate for mandatory booster vaccination rate, $F_{20} = 0.002$, compared to the mandatory primary series vaccination rate. We iteratively increased the mandatory testing rates, $T_{10}$ and $T_{20}$ starting from 0.0025 (average proportion of reported daily tested people prior to February 01, 2022), with the intention of including the rate for people

**Table 2. Parameter's baseline values. CI represents confidence interval.**

| Parameter | Baseline value | Parameter range | Reference |
|---|---|---|---|
| $\pi$ | 1710 indivi./day | [1282, 2137] | See Sec. (*i*) |
| $\mu$ | $3.3 \times 10^{-5}$ per capita per day | $[2.47 \times 10^{-5}, 4.12 \times 10^{-5}]$ | See Sec. (*i*) |
| $\epsilon$ | 0.33 $day^{-1}$ | [0.25,0.41] | [32] |
| $\tau$ | 0.8 | [0.6, 0.95] | [33] |
| $\rho$ | 1/14 $day^{-1}$ | [1/11, 1/19] | [32] |
| $\rho_h$ | 1/10 $day^{-1}$ | [1/8, 1/13] | Assumed ($\rho_h < \rho$) |
| $\eta_1$ | 0.71 | 95% CI: (0.62, 0.8) | [20] |
| $\eta_2$ | 0.41 | 95% CI: (0.32, 0.53) | [19] |
| $\eta_3$ | 0.46 | 95% CI: (0.36, 0.57) | [19] |
| $\eta_4$ | 0.85 | 95% CI: (0.29, 0.98) | [19] |
| $\phi$ | 1/180 $day^{-1}$ | [1/161, 1/270] | [19] |
| $h_1$ | 0.0012 $day^{-1}$ | [0.0009,0.0015] | [34] |
| $h_2$ | 0.000312 $day^{-1}$ | [0.000234,0.00039] | [19] |
| $F_{max}$ | 0.027 $day^{-1}$ | [0.02, 0.034] | See Sec. (*i*) |
| $F_{10}$ | 0.0023 $day^{-1}$ | [0.0017, 0.003] | See Sec. (*i*) |
| $F_{20}$ | 0.002 $day^{-1}$ | [0.0015, 0.0025] | See Sec. (*i*) |
| $T_{max}$ | 0.8 | [0.6, 0.95] | See Sec. (*i*) |
| $T_{10}$ | 0.2 $day^{-1}$ | [0.15, 0.25] | See Sec. (*i*) |
| $T_{20}$ | 0.2 $day^{-1}$ | [0.15, 0.25] | See Sec. (*i*) |
| $1/a$ | 3 days | [2,4] | [35] |
| $D$ | 50 | [37, 63] | See Sec. (*i*) |
| $B$ | 100 | [75, 125] | See Sec. (*i*) |
| $\theta$ | 0.5 | [0.37, 0.62] | Assumed |
| $\delta$ | 80% | [0.4, 0.96] | Assumed |
| $\alpha_i, i = 1, 2$ | 0.5 | [0.25,0.75] | Assumed |
| $d_1$ | 0.000034 $day^{-1}$ | [0.000025, 0.00004] | See Sec. (*i*) |
| $d_2$ | 0.00031 $day^{-1}$ | [0.00023, 0.00039] | See Sec. (*i*) |
| $\beta_1$ | 0.17 $day^{-1}$ | 95%CI: (0.056, 0.29) | Fitted |
| $\beta_2$ | 0.66 $day^{-1}$ | 95%CI: (0.615, 0.702) | Fitted |
| $k$ | 0.52 | 95%CI: (0.437, 0.607) | Fitted |
| $\xi$ | 0.99 | 95%CI: (0.741, 1.258) | Fitted |
| $1/a_1$ | 2 days | [1,3] | Assumed |
| S2(0)+S3(0) | 44,365,186 (85% total population) | [70%, 98%] | [36] |
| E1(0)+E2(0) | 405,339 | [283,737, 689,076] | |

who can undergo testing at home, to achieve a best fit to the data. The process resulted in $T_{10} = 0.2$, $T_{20} = 0.2$. Assuming that a substantial number of people can be tested (both at home and in health centers) than vaccinated during the omicron time, we fixed the value of $T_{max}$ at 0.8. This value ($T_{max} = 0.8$) represents a theoretical upper bound for the testing rate, which is assumed to be achievable during periods of high information coverage—when the population is highly concerned about the epidemic. While practically unattainable, we adopt this theoretical maximum to facilitate model fitting. In the model, this number is equivalent to testing 80% of infectious individuals. At the beginning of a vaccination campaign, the cumulative number of vaccinated individuals is assumed to grow linearly. The reactivity factor to prevalence ($D$) was selected to produce an initial linear increase

in vaccination rate. This was done by iteratively plotting the vaccination rate for various randomly chosen values of $D$ within the range [1, 100], a range informed by prior studies—ranging from small values like 2.5 [16] to larger ones like 6,500 [14]. Based on this process, we chose $D=50$. A similar approach was assumed for reactivity to prevalence in testing rates. Given that perceived severity tends to outweigh perceived risk of infection [38], we assumed that reactivity to severity information is twice that of prevalence, and set $B=100$. This heightened reactivity reflects the fact that severity-related events are more publicly visible and emotionally impactful, often triggering stronger behavioral responses. Given the lower severity associated with the Omicron variant, the death rates ($d_1$ and $d_2$) parameter was initialized at a small value and iteratively adjusted during the fitting procedure. At each iteration, we evaluated the agreement between the model output and the observed mortality data, selecting the value that provided the best overall fit. The initial conditions for partially immune susceptible population were informed by a seroprevalence data reporting adjusted prevalence rates of 98.5% for anti-spike antibodies and 70% for anti-nucleocapsid antibodies [36], indicating that the majority of the South Korean population had acquired immunity through vaccination and/or prior infection during this period. Guided by these findings, we assumed that 85% of the total population (44,365,186 out of 51,815,808) possessed partial immunity subject to waning at the start of the simulation on February 1, 2022, and were therefore assigned to the partially immune susceptible compartments. Consistent with the lower prevalence of infection-only immunity [36], we further assumed that approximately 90% of partially immune individuals belonged to the vaccination-derived immunity class ($S_2$) and 10% to the infection-derived immunity class ($S_3$), yielding $S_2(0)=40,000,000$ and $S_3(0)= 4, 365, 186$. Based on the initial population distribution (85% in secondary dynamics and 15% primary dynamics), we used the same distribution for the initial conditions for infected and vaccinated compartments in both dynamics. For example, from the data, the number of new vaccinated people on February 01, 2022 was 3,323. Therefore, 85% of these people are in $V_2$ (most of this is booster vaccination), and the remaining are in $V_1$. That is, $V_1(0) = 431$ and $V_2(0) = 2891$. We assumed a 10% (half of the baseline mandatory testing rate) of the daily cases for the initial conditions for tested compartments, thus, $I_{T1}(0) = 2634$ and $I_{T2}(0) = 2026$. We assumed no undetected cases at the beginning of the simulation, so that $I_{U1}(0) = I_{U2}(0) = 0$. Following the modeling approach of [39], we assumed that the number of exposed individuals is approximately 20 times the number of symptomatic cases. We note that this is a simplifying assumption commonly used in early outbreak modeling studies to initialize the exposed population, and the exact ratio is uncertain. We adopted a larger initial exposed population to account for the possibility of substantial undetected or pre-symptomatic infections. Because the model is initialized with zero undetected infectious individuals, this higher exposed count transitions into the undetected infectious compartment during the early phase of the simulation. Under this assumption, the initial exposed population is estimated to be 405,339 which is distributed as $E_1(0)=52,694$ and $E_2(0)=352,645$. A one-way sensitivity analysis result showed that the variation (70% to 95% from 405,339) in the initial exposed population has minimal effect on the cumulative incidence. We assumed a 10% of the exposed people are initially symptomatic and 20% are asymptomatic. Therefore, we set $A_1(0) = 0.2 * E_1(0) = 10, 538, I_1(0) = 0.1 * E_1(0) = 5269$ and $A_2(0) = 0.2 * E_2(0) = 70529, I_2(0) = 0.1 * E_2(0) = 35264$. From Our World in Data, we used the number of people in the ICU on February 01, 2022 as an estimate for the initial population in the hospital; therefore, $H(0)= 203$. The initial condition for the death compartment is, $D(0) = 15$, that is, the number of dead individuals on February 01, 2022. We assumed recovered individuals to be $R(0)=200$, more than 10 times the number of deaths. This reflects that many more people recover from the disease after experiencing mild illness due to their immunity. Finally, the rest of the population is placed in the $S_1$ compartment. *i.e.* $S_1(0) = N(0) - \sum_i Z_i(0)$, where $Z_i$ represents all compartments in the model except $S_1$ and $D$. As for the information indices, we set their initial conditions at the equilibrium [16]: *i.e.,* $\mathcal{V}(0) = k(I_{T1}(0) + I_{T2}(0) + H(0))/N_0$ and $\mathcal{N}(0) = k(d_1(I_{U1}(0) + I_{U2}(0)) + d_2 H(0) + H(0))/N_0$.

*(ii) Model fitting with South Korea COVID-19 data*

We fitted the model (10) to the cumulative daily cases, daily vaccination, and daily death data for the time period from February 01,2022 to May 31,2022. We fit our model and conducted numerical simulations to the highest peak of the omicron wave to evaluate how intervention scenarios and information parameters can influence the peak of the epidemic. The fitting process is accomplished by a Python built-in curve fitting function called *curve_fit* (Nonlinear least squares

optimization method) [40]. In general, this method identifies the best parameter values by minimizing the sum of squared errors between the model output and the data sets. For our model, there are four parameters to be estimated: transmission rates in primary and secondary dynamics ($\beta_1$ and $\beta_2$), information coverage ($k$), and testing modification of asymptomatic individuals ($\xi$). In addition to estimating these optimal values, we computed 95% confidence intervals using the parameter covariance matrix returned by the nonlinear least-squares procedure, providing a quantitative measure of uncertainty around each fitted parameter. To evaluate model performance across the three observational data streams, we also computed the min–max normalized RMSE [41], which provides a scale-independent measure of goodness of fit and enables direct comparison among datasets of markedly different magnitudes (cumulative cases and vaccinations in millions, versus deaths in thousands). The resulting normalized RMSE values were 0.022 for cumulative cases, 0.091 for vaccinations, and 0.015 for deaths. These low normalized errors indicate that the model captures the temporal dynamics of all three datasets with strong relative accuracy, despite substantial differences in their absolute scales.

Visually, the model captures the overall trajectory of both series, with slightly closer alignment for cumulative deaths and minor under- or over-estimation in the cumulative case series See Fig 3 panels a and c. The cumulative vaccination data follows some irregular patterns as compared to cases and death data and the model approximation changes over time (it slightly underestimates, or overestimates), 3 panel b. The estimated parameters, together with all fixed model parameters, are summarized in Table 2. For parameters obtained from the literature, we report the corresponding 95% confidence intervals when available. For parameters estimated through model fitting, the table includes the 95% confidence intervals derived from the estimation procedure. In addition, for parameters included in the one-way sensitivity analysis (Section (v)), we present the minimum and maximum values defined as ±25% of the baseline value.

In Fig 4, we show the time series of observed daily COVID-19 cases alongside the model-generated daily number of detected cases (incoming to $I_{T1} + I_{T2}$, purple curve). These daily values correspond to the data that were cumulatively aggregated for the model fitting shown in Fig 3. Although the calibration was performed using cumulative reported cases,

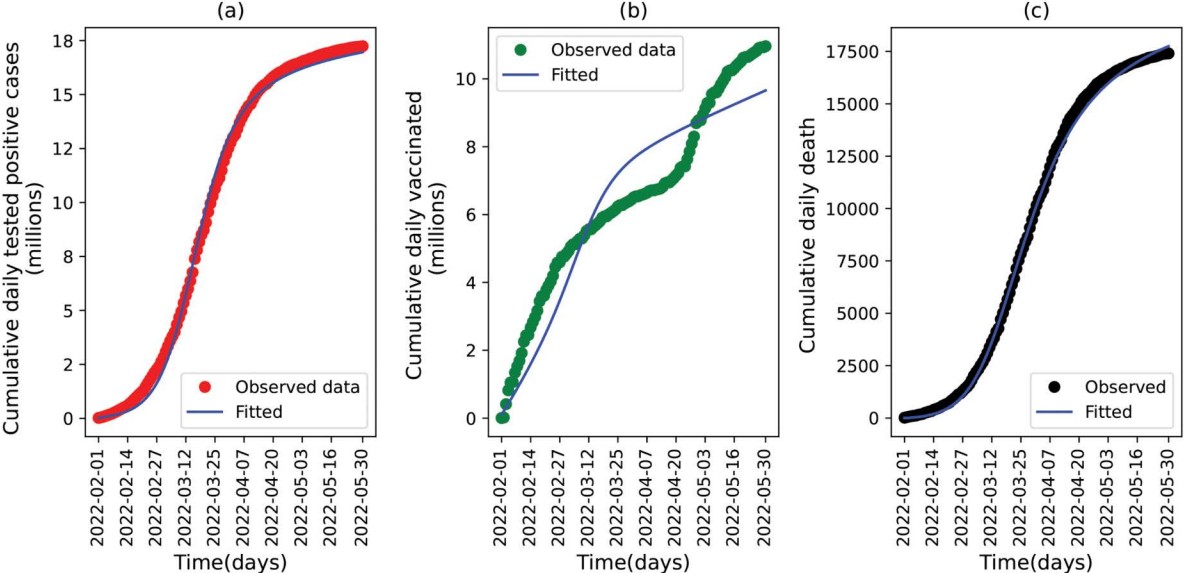

**Fig 3. Result of the model fitting to South Korea's COVID-19 epidemic data.** The data represents the cumulative observed daily cases (in panel a), vaccination (in panel b) and death (in panel c). The time frame is from February 01, 2022 to May 31, 2022. The blue curve shows the approximation by the model (10). The approximation for a number of cases is obtained by adding a number of individuals in $I_{T1}$ and $I_{T2}$ classes (detected cases). Similarly, the estimated number of vaccinations is obtained by summing the number of individuals in $V_1$ and $V_2$ classes. The initial conditions used for daily cases, daily vaccination and daily death are $V_1(0) + V_2(0) = 3322$, $I_{T1}(0) + I_{T1}(0) = 4660$ and $D(0)=15$, respectively.

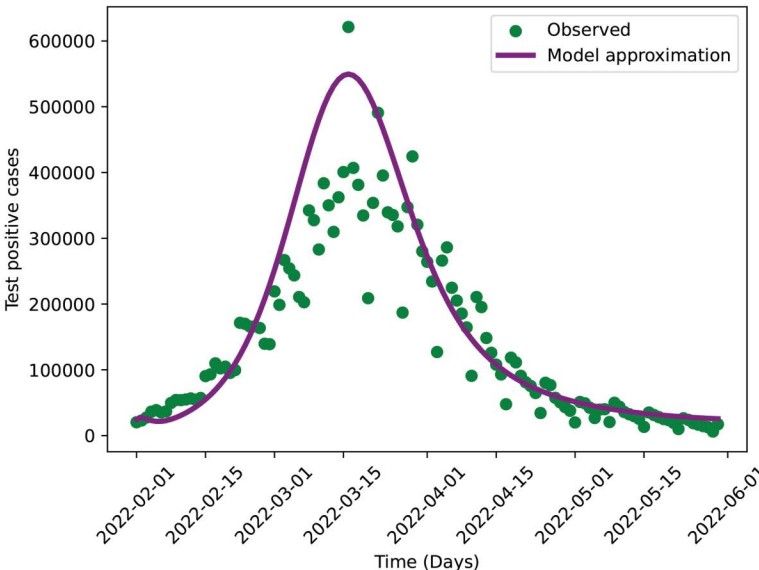

**Fig 4. A comparison of observed daily cases (green scattered plot) with model approximation of daily cases (purple curve).**

this figure provides an additional validation by demonstrating how well the model replicates the temporal pattern of daily incidence. As shown, the model captures the overall trend and variability of the daily reported cases, indicating consistency between the cumulative fit and the underlying daily case dynamics.

Using the parameter values listed in Table 2, the estimated effective reproduction number is $R_e = 1.24$. This estimate is comparable to the value reported by [42], who found the reproduction number to be 1.3 for the Omicron variant during the period from November 25, 2021 to January 8, 2022. According to Theorem 1, the disease-free equilibrium is unstable under the baseline parameter values. When the model is simulated over a longer time horizon, the dynamics for infected compartment approaches towards an endemic equilibrium, see Fig 2 panel b. The initial rise followed by a decline in incidence observed over the study period is likely attributable to the effects of vaccination and enhanced testing rates.

*(iii) Sensitivity analysis*

We conducted a comprehensive one-way sensitivity analysis to quantify the influence of individual model parameters on the cumulative incidence of COVID-19 over a four-month period (February 1, 2022–May 31, 2022). For each parameter, the cumulative incidence was recomputed while varying that parameter individually and holding all others fixed at their baseline values. The results are summarized using a tornado plot, which visually ranks parameters according to their relative impact on the outcome. For clarity, only the top 15 most influential parameters are presented in Fig 8. In addition to epidemiological and behavioral parameters, we included selected initial conditions—specifically, the size of the partially immune susceptible population and the number of exposed individuals at the start of the simulation. These quantities are subject to substantial uncertainty: the initial number of exposed individuals, which is not directly observed, was inferred using an assumed scaling of reported symptomatic cases to represent unobserved infections, while the partially immune population may vary due to immune waning, heterogeneous immune responses, and demographic factors such as age structure. For parameters with available empirical evidence or those estimated through model fitting, lower and upper bounds were defined using the corresponding 95% confidence intervals. For parameters lacking reliable data, we assumed an uncertainty range of$\pm$25% around the baseline value. In contrast, initial-condition parameters were allowed to vary within 70% to 98% of their baseline values to reflect plausible uncertainty in population immunity and latent

infections. The complete set of parameter ranges used in the sensitivity analysis is summarized in Table 2. In the tornado plot, the effect of parameter uncertainty on cumulative incidence is represented by horizontal bars extending from the baseline (center) value. The length of each bar reflects the change in cumulative incidence associated with the low and high values of the corresponding parameter, thereby providing a clear visual comparison of their relative importance.

## Scenarios

The compliance of individuals with protective measures may vary over the course of a pandemic [43,44], especially in scenarios where the disease persists despite widespread vaccination, as seen in the case of COVID-19. In such circumstances, it becomes essential to analyze the dynamics by accounting for behavioral disparities among sub-populations and their different risk perceptions. We accounted for the disease status information (prevalence and severity) that can influence the sub-populations' (susceptible and partially immune) risk perception and compliance with vaccination and testing (i.e., susceptible people may prioritize prevalence while partially immune people may prioritize the severity (representing a change in risk perception [44]), and this phenomenon will impact their compliance to vaccination and testing differently). We assessed the consequence of vaccination and testing rates and the epidemic burden (number of active cases) by applying different weights (using $\alpha_1$ and $\alpha_2$ for prevalence and $1 - \alpha_1$ and $1 - \alpha_2$ for severity) between two information (prevalence and severity) among the susceptible and partially immune populations. We examine the following three scenarios to demonstrate the impact of the weight individuals place on the information about prevalence or severity in the two dynamics, i.e., which information people care about when making a decision to be tested or vaccinated:

1. **Scenario 1 (base case)**: Susceptible (in primary dynamics) and partially immune (in secondary dynamics) people equally care about both severity and prevalence, given by $\alpha_1$ = 0.5, $\alpha_2$ = 0.5 (the base case) for taking a decision to vaccinate or test regardless of their immune status.

2. **Scenario 2**: Both susceptible and partially immune people prioritize prevalence information, given by $\alpha_1$ = 0.9, $\alpha_2$ = 0.9.

3. **Scenario 3**: Susceptible people prioritize prevalence information and partially immune people prioritize severity, given by $\alpha_1$ = 0.9, $\alpha_2$ = 0.1.

We set the above scenarios based on theoretical assumptions due to lack of data regarding how people prioritize information for decisions.

## Results

In this section, we perform numerical simulations to investigate the effect of parameters related to voluntary vaccination and testing on the disease dynamics. Except for the varying parameters in the plots, all other parameter values are fixed as in Table 2.

*(i) Impact of changes in information parameters on active cases peak*

Here, we address the effect of changes in information coverage, information delay and reactivity factors on active cases peak. Active cases indicate the number of infectious individuals that are not detected (tested), those in $I_{U1}$ and $I_{U2}$. Fig 5 illustrates that information coverage is the key driver among other parameters. The maximum of active cases is more sensitive to the information delay when the information coverage is larger than the base line value (>50%). The minimum peak of active cases is attained when there is a higher information coverage ($k > 80\%$) and shorter time delay ($T \leq 3$ days), see Fig 5 panel (a). For example, when $k = 0.9$ and $T = 3$ days the peak of active cases is around 8 million, whereas when $k = 0.1$ and $T = 30$ days the peak is increased to 12.5 million. The results shown in Fig 5, panels (b) and (c), indicate that although individuals respond to severity information at twice the rate of their response to prevalence information, the peak of active cases is more sensitive to their reactivity to prevalence than to severity. This heightened sensitivity to

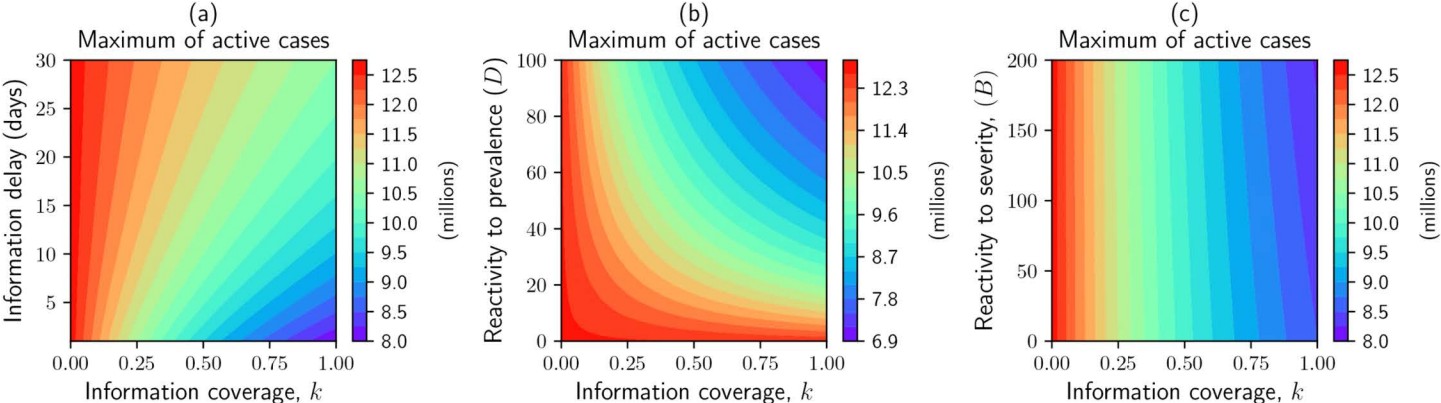

**Fig 5. Contour plot for maximum (peak) of active cases by varying information coverage (_k_) with: information delay time (1/_a_ days), panel (a), people's reactivity to prevalence information (_D_), panel (b), reactivity to severity information (_B_), panel (c).** Active cases represent a number of infectious individuals who are undetected ($I_{U1} + I_{U2}$).

prevalence-related information can be attributed to the stronger signal generated by the prevalence indicators (i.e., number of tested and hospitalized individuals), which are generally higher in magnitude compared to the indicators of severity (i.e., number of deaths and hospitalizations). When people reaction to the prevalence, _D_, is less than 20; increasing information coverage does not affect the active cases peak, Fig 5 panel (b). This demonstrates that prompt public response is an essential factor in addition to the government's efforts to achieve high information coverage in a timely way to reduce the epidemic peak.

*(ii) Impact of change in level of reactivity to the information by partially immune population on the dynamics of active cases*

Here, we examine how relative responsiveness to prevalence/severity information by partially immune individuals compared to susceptible individuals, represented by $\theta$, affects the trajectory of active cases. A value $\theta = 1$ indicates equal reactivity to information as susceptible individuals, whereas $\theta = 0$ denotes no reactivity. At the base line value $\theta = 0.5$ (50% reduction), the peak of active cases becomes 9.7 million, which we refer to as the base line scenario. The lowest peak is obtained when partially immune individuals respond to information at the same level as susceptible individuals ($\theta = 1$), resulting in a 15% reduction relative to the baseline, and a 33% reduction compared to the scenario where partially immune individuals do not react to information at all ($\theta = 0$), see Fig 6. For an intermediate value $\theta = 0.75$ (25% reduced reactivity), the peak decreases by 8% relative to the baseline. In general, the active cases peak lowers as partially immune people behave more similarly to non-immune persons (*i.e.*, $\theta$ changes from 0 to 1).

*(iii) Impact of behavior adaptation between immune and susceptible populations on the dynamics of vaccination and testing rates and active cases.*

In this subsection, we examine the three scenarios defined in the Method section that represent how behavioral adaptation—captured through changes in risk perception related to infection and disease severity—influences care-seeking behaviors (vaccination and testing) and, in turn, affects the resulting disease dynamics (active case trajectories). The results of scenario 3 (Fig 7 panel (c)) show that vaccination and testing rates in primary dynamics are the highest, while the vaccination and testing rates of the partially immune population are the lowest. As susceptible populations rapidly become partially immune populations with the model simulations over time, the lower vaccination and testing rates of partially immune populations (second dynamics) resulted in a higher (20%) peak of active cases compared to the base case (scenario 1, Fig 7 panel (a)). On the other hand, the results of scenario 2 (Fig 7 panel (b)) show that the vaccination and testing rates of both susceptible and

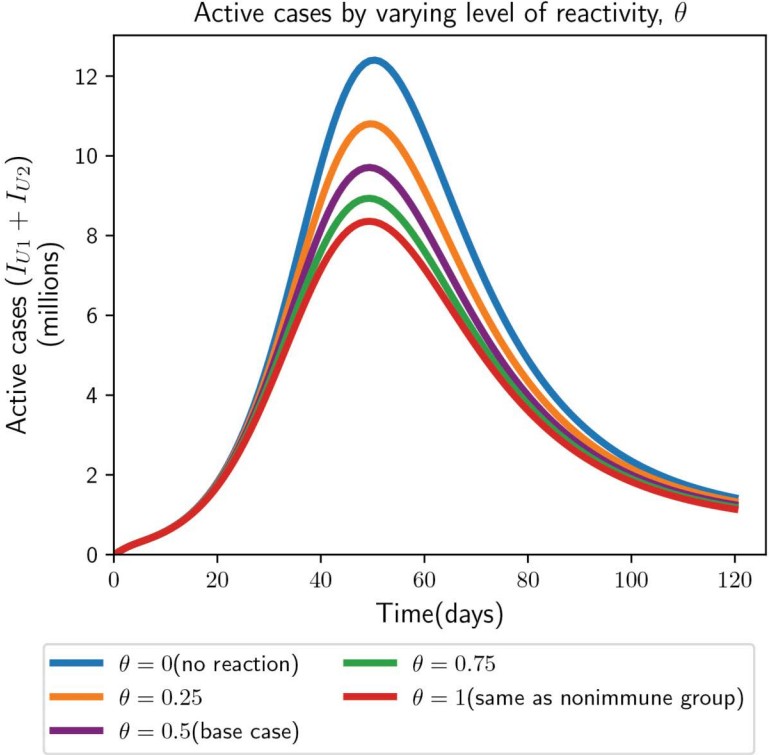

**Fig 6. Time series of active cases, with varying levels of reactivity to the information by partially immune people (in secondary dynamics) relative to susceptible (people in primary dynamics).** The baseline value of the level reactivity to information by non-immune individuals is $D = 50$ for prevalence and $B = 100$ for severity.

partially immune populations are slightly higher than the base case, as they react more from the prevalence signal to vaccination and testing than base case (90% vs 50%), resulting in a lower (24%) peak of the active case than the base case. A shift in prioritization from prevalence to severity information among partially immune individuals (from 90% prevalence and 10% severity to 90% severity and 10% prevalence) while susceptible individuals remain at the same risk perception (90% prevalence and 10% severity) can result in an increase of active cases peak by 57% (from 7.4 million to 11.6 million) compared to the case where both (partially immune and susceptible individuals) have higher weight of prevalence than severity (90% prevalence and 10% severity), comparing Fig 7 panels (b) and (c) (shift from scenario 2 to scenario 3).

The numerical simulation results suggest that differences in risk perceptions among sub-populations may result in different voluntary decisions to vaccination and testing and levels of prevalence may peak consequently. Therefore, the overall epidemic burden is the result of the relative size and distribution of sub-populations and their different risk perceptions and associated care-seeking behaviors. A rise in epidemic can lead to increased voluntary care-seeking behaviors, which can reduce the prevalence. This reduced prevalence can decrease voluntary care-seeking behavior, which in turn can increase the prevalence. Such feedback loop in the model (10) can help us understand a dynamic interplay between disease status and population behaviors. Future work should explore fitting the proposed model to the course of epidemic waves with the seroprevalence and care seeking data from survey, where participants are queried in different time periods of the pandemic about their immune status, risk perception, and contact patterns or care seeking (vaccination and testing) behaviors. Such data can be used to improve the estimates of some of the parameters in the behavior adaptation metrics introduced in this paper.

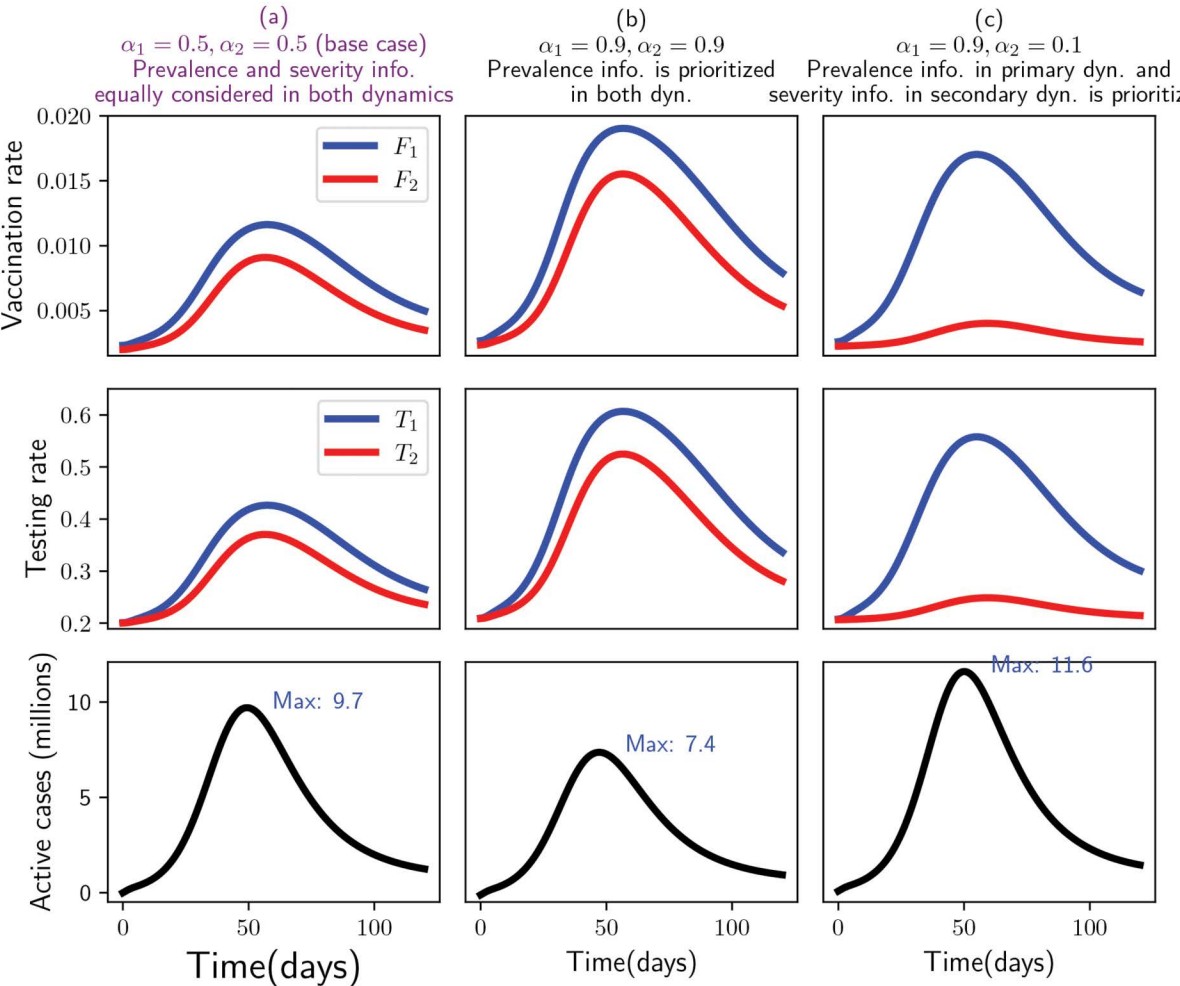

**Fig 7. The dynamics of vaccination rate (in primary dynamics $F_1$, in secondary dynamics $F_2$), first row, Testing rate (in primary dynamics $T_1$, in secondary dynamics $T_2$), second row, and Active cases ($I_{U1} + I_{U2}$), (third row) under three scenarios: first, people in both dynamics equally care about information regarding prevalence and severity, panel (a), second, people in both dynamics care about prevalence than severity information, panel (b), third, people in primary dynamics care about prevalence than severity whereas people in secondary dynamics care about severity than prevalence, panel (c).** The parameters $\alpha_1$ and $\alpha_2$ indicates the weight given to the prevalence information in primary and secondary dynamics respectively. The remaining (complementary) weights $1 - \alpha_1$ and $1 - \alpha_2$ are assigned to severity information in the respective dynamics.

*(v) Sensitivity of cumulative incidence to parameters.*

The results of the one-way sensitivity analysis, summarized in Fig 8, indicate that cumulative COVID-19 incidence is most strongly influenced by booster vaccination effectiveness, the recovery rate of symptomatic non-hospitalized individuals, and the progression rate from primary to secondary infection dynamics. These parameters emerge as the dominant drivers of model outcomes across the full parameter set considered. Although the top ten most influential parameters are predominantly classical(non-behavioral) parameters, behavioral parameters—including information prioritization among immune individuals, reduced behavioral reactivity in partially immune populations, and reactivity to prevalence-based information—also exert a non-negligible influence, ranking between 12th and 14th in terms of impact. Overall, parameters associated with secondary infection dynamics (e.g., booster vaccination effectiveness $\eta_4$, prevalence prioritization among partially immune individuals $\alpha_2$) tend to exhibit greater influence on cumulative incidence than their counterparts in

**Fig 8. Tornado plot showing the sensitivity of parameters to cumulative incidence over four months (February 01, 2022 – May 31, 2022).** The red and blue bars show the cumulative incidence corresponding to a low and high values of the parameters, respectively. The center vertical line represents the cumulative incidence when all parameters are fixed at their baseline values. The numbers in the closed bracket for each parameters show the low and high values used for sensitivity analysis: [low, high].

the primary dynamics, although notable exceptions arise when primary-dynamics parameters are characterized by wider uncertainty ranges. For example, the transmission rate in the primary dynamics ($\beta_1$) shows a larger sensitivity effect than its secondary-dynamics counterpart ($\beta_2$), which is attributable to the wider 95% confidence interval for $\beta_1$ rather than to a greater intrinsic sensitivity of the model. The heightened sensitivity to parameters in secondary dynamics reflects both the relatively large fraction of the population initially classified within secondary dynamics and the rapid transition from primary to secondary states embedded in the model structure. By contrast, the model exhibits relatively low sensitivity to uncertainty in the initial number of exposed individuals, which ranks 19[nd] in influence. In comparison, variation in the initial size of the partially immune susceptible population has a more pronounced effect on cumulative incidence. Together, these findings highlight the central role of immunity-related processes—particularly those governing booster effectiveness and secondary infection dynamics—in shaping epidemic outcomes during the study period.

## Discussion

In this study, we developed a novel behavior-epidemiology model representing the transmission of COVID-19 that takes into account the behavioral differences among people who are partially immune and those who are not-immune (susceptible) in seeking vaccination and testing [43–45]. The model outputs were fitted to observed cumulative COVID-19 cases, vaccination and mortality data during the Omicron wave (February 01, 2022 – May 31, 2022) in South Korea. Unlike other behavioral models that employ the information index approach with a single type of information such as disease prevalence [14,16,35,46,47], our model takes into account that people may have different risk perception (prevalence and severity) and behavior responses (vaccination and testing) across different subgroups by immune status. The overall impact on the prevalence peak in our behavioral model is the result from the non-linear relationship between the number of detected cases that can be information signals to promote voluntary vaccination and testing (decreasing prevalence) and the number of undetected cases that can keep contributing to the transmission (increasing prevalence). These results should be interpreted in the context of the Omicron wave, when the population immunity was high and perceived infection risk was comparatively low. In such settings, individuals may discount prevalence information and give greater attention

to severity signals, yet prevalence still shapes protective behaviors. This underscores that even in highly immune populations, dynamic shifts in risk perception—and the resulting behavioral responses—remain central drivers of transmission patterns.

Our stability analysis shows that if mandatory testing rate for immune individuals is decreased below or increased above the threshold values ($T_{20}$ = 0.4 as shown in the yellow frontier line in Fig 2), $R_e$ can be greater than or less than one. The impact of voluntary behavior parameters become more important when the mandatory vaccination or testing rates are unable to reduce the reproduction number below 1. For example, with $R_e$ = 1.24 (a value obtained with baseline parameter values), the cumulative active (undetected) cases become 36.8 million. This cumulative undetected cases can be reduced by up to 55% by enhancing the voluntary vaccination and testing, which can be achieved by increasing the behavior parameters related to the voluntary vaccination and testing rates from their baseline value. For example, at baseline, partially immune individuals respond less strongly (50%) to prevalence information than susceptible individuals. Increasing their reactivity to match that of susceptible individuals, cumulative undetected cases can be reduced by 10% (33.2 million). In addition to this increment, if we increase all other behavioral parameters related to voluntary vaccination and testing rates to some possible maximum value (information coverage (baseline-0.52 to 0.95), prevalence prioritization (baseline-0.5 to 0.95), reactivity to prevalence information (baseline-50–100) the cumulative undetected cases can be reduced by 55% (16.7 million). These illustrative calculations are provided for explanation and are not included in the manuscript figures or main results. Note that the effects of increasing voluntary vaccination and testing reported here assume fixed mandatory vaccination and testing rates. Uncertainty of these behavioral parameters is incorporated through both direct parameter variation (Figs 5, 6) and the one-way sensitivity analysis (Fig 8).

Our simulation results indicate that, under fixed mandatory vaccination and testing rates, increasing the reactivity of partially immune individuals to match that of susceptible individuals can reduce the peak number of active cases by up to approximately 33% compared to the scenario in which partially immune individuals do not react to information. This suggests that the response to protective measures by individual immune status can substantially influence the level of future incidence. The result related to information prioritization by susceptible and partially immune individuals showed that over time, if partially immune individuals prioritize severity over prevalence (altering their risk perception), the peak of active cases can increase by up to 20% and 57% compared to when they equally (50% to prevalence and 50% to severity) prioritize both information (base case) and when they prioritize prevalence over severity (90% to prevalence, 10% to severity), respectively. This result shows that when public risk perception changes (e.g., toward severity over prevalence) over the course of the pandemic with multiple variants of epidemic waves, the required target of mandatory vaccination or testing may change, and the public health efforts to control disease may require additional endeavors to reduce the prevalence peak. Furthermore, the variation in prevalence peak based on the risk perception by susceptible and partially immune people underscore the need for further research into how these groups of people perceive disease-related information. While our model is calibrated to the Omicron wave, the qualitative insights regarding risk perception, voluntary vaccination, and testing remain applicable to future waves characterized by high transmissibility and partial immune escape. However, in a steady endemic phase—where immunity stabilizes and behavioral responses become less volatile—the quantitative impact of information-driven behavior may be smaller. Thus, the results are most directly applicable during epidemic waves or periods of rapid epidemiological surge rather than long-term endemic phase.

The sensitivity analysis results show that the key drivers influencing the cumulative incidence of undetected infections are booster vaccination effectiveness, the recovery rate of symptomatic non-hospitalized individuals, and the progression rate from primary to secondary infection dynamics. This highlights the central role of booster-induced protection and immunity waning in shaping epidemic trajectories during the study period. In the model, individuals transition from primary to secondary dynamics approximately six months after completion of the primary vaccination series or prior infection, reflecting waning immunity over time. Consequently, parameters governing this transition and the effectiveness of booster vaccination exert a disproportionate influence on cumulative incidence, as they directly modulate the size and

susceptibility of the partially immune population. Although classical epidemiological and immunity-related parameters dominate the top ranks of the sensitivity analysis, behavioral parameters also exert a meaningful influence on cumulative incidence. In particular, parameters governing information prioritization, responsiveness to prevalence signals, and reduced behavioral reactivity among partially immune individuals appear among the influential drivers. The non-negligible sensitivity to behavioral parameters highlights the continued relevance of risk communication and information dissemination strategies, even in settings with high levels of vaccination-induced or infection-acquired immunity. As population immunity wanes and individuals transition into secondary dynamics, behavioral fatigue or reduced risk perception among partially immune groups may offset gains achieved through vaccination alone. These results therefore suggest that interventions targeting behavioral responses—such as timely, targeted communication emphasizing ongoing risk and the benefits of protective behaviors—can complement biomedical interventions and enhance epidemic control, particularly during periods of immune waning and booster rollout.

There may be various other metrics that could induce behavior changes during a pandemic like COVID-19. For example, authors in a recent study [7] developed a behavioral epidemiology model aiming at assessing the impact of human behavior changes due to factors such as disease-related information received from members of the other group, the level of symptomatic transmission in the community, the proportion of non-symptomatic individuals in the community, the level of publicly available disease-induced mortality information, and fatigue with adherence to control and mitigation interventions in the community, on compliance behavior. One of the findings in their study is that disease-induced mortality has a greater influence on behavioral change than the level of symptomatic transmission. This contrasts with our model results, which suggest that the prevalence of symptomatic infection—reflected in the peak of active cases—is a more significant driver of behavioral response than disease severity, as measured by deaths and hospitalizations. This discrepancy may be explained by two main factors. First, in our model, information about disease prevalence is inferred from the number of individuals tested and hospitalized, which typically exceeds the number of deaths or severe cases. Consequently, the informational signal triggering voluntary vaccination and testing behavior is stronger for prevalence than for severity. Second, the fact that their study examined the impact of each different model output as described above as a signal to change adherence behaviors among the susceptible population in the early stage of the pandemic (thus, most people are susceptible yet) and fit the model to the relatively high mortality rate under the original SARS-CoV-2 virus variant compared to the case during Omicron variants, which resulted in mortality being a driving factor of adherence behavior. In our study, we accounted for behavior adaptation by the individual history of previous exposure and vaccination and explored differing individual reactivity and risk perception/weight by the type of information (between prevalence and severity) and immune status and fitted the model to data during the Omicron epidemic peak (of which the virus variants are characterized by high transmission but low severity [48]). Overall, these findings can be complementary to understand a population behavior dynamic in the early stage of the pandemic, where the adherence behavior of susceptible populations can be a function of various aspects of model outputs, and in the later stage of a pandemic, where care seeking behavior can differ by population immunity status and their risk perception.

Our study has some limitations. First, empirical data are scarce for validating the information index functions (Eqns. (6)-(9)) that drive vaccination and testing uptake. We therefore (i)- iteratively calibrated key parameters, especially reactivity to prevalence information – so modeled uptake mirrored the slow early growth seen in real campaigns [31] and (ii)- performed extensive sensitivity analyses to ensure findings remained stable across plausible parameter ranges. Second, our behavioral response functions $g_1$ (tested and hospitalized cases) and $g_2$ (driven by only new deaths and hospitalizations)-skew attention towards prevalence and may down weight the impact of severe outcomes. To address this bias, we assumed a twofold greater reactivity to severity information compared to prevalence. This adjustment reflects the notion that severity-related events are generally more visible to the public and psychologically impactful, often prompting stronger behavioral responses—despite the fact that the twofold assumption remains theoretical. Future work should aim to develop behavior rate formulations grounded in empirical data. In particular, both message functions should be

calibrated against observed data on public perception to improve the realism and accuracy of the behavioral assumptions. Third, our model accounts for the driving forces for vaccination and testing decisions driven by individuals' risk perceptions about the disease status in the population. However, other factors can influence testing and vaccination decisions, such as access to health care systems, testing procedures, perceptions or beliefs related to vaccine side effects, knowledge of COVID-19 symptoms, etc. [49–51]. Furthermore, it is also possible that the individual reactivity to prevalence and severity may differ by other individual characteristics such as age group as well (besides the immune status); in other words, in communities where the majority of people are young, severity information may not be as concerning since COVID-19 deaths predominantly affect older individuals [52]. Conversely, if the majority of the population is old, the average weight of risk perception in the population toward death may be greater than the infection. Indeed, it may not be possible to parameterize all the possible factors and individual heterogeneity that influence willingness to testing/vaccination and behaviors in a single model. Fourth, although our model incorporates the effectiveness of vaccine-induced immunity against infection, it does not account for the time delay between vaccine administration and the onset of protective immunity. This simplification may lead to an overestimation of the immediate impact of vaccination on disease transmission dynamics. A further limitation of this study is that our analysis focused primarily on numerical simulations due to the high dimensionality and nonlinear structure of the model, which makes full analytical treatment difficult. As a result, we did not investigate deeper analytical properties such as bifurcation structure or chaos-like behavior. Advanced methods used in recent nonlinear dynamics studies [53–56] such as phase–portrait methods and bifurcation techniques could be applied in future research to further characterize model behavior near the threshold $R_e = 1$ and to explore additional qualitative dynamics. Finally, while our study captures individual behavior during an epidemic wave, it does not explicitly model long-term endemic equilibrium or waning behavioral responsiveness over time. Therefore, care should be taken when extrapolating short-term behavior-driven effects to post-peak or endemic periods.

Decision-makers are faced with the daunting task of interpreting model predictions while simultaneously estimating how behavioral responses should alter predictions. Despite the given complexities and uncertainties, these estimates might be enhanced by explicitly modeling behavioral responses (i.e., willingness to vaccination as a function of risk and benefit of vaccination) to interventions rather than simply adjusting any assumed constant parameters. Accordingly, our study findings demonstrate the relatively significant impact of behavioral factors on the overall cumulative incidences and thus emphasize the importance of future efforts to collect empiric data and identify driving factors on risk perception and information prioritization among immune and susceptible individuals. Incorporating these behavioral aspects into transmission models may become increasingly useful and important as the epidemic continues and people's behaviors change.

## Conclusion

This study contributes to the field of epidemiological modeling by illustrating the complex interplay between information, human behavior, and immune status and their impact on disease transmission. To our knowledge, our study is the first attempt to apply an information index with different disease-relevant information (prevalence and severity) among people with different immune statuses and to fit such a model to real-world data. Future studies may further evaluate the interplay between various individual status (immunity/age/care seeking history), vaccination, testing and disease transmission, aiming at providing optimal intervention strategies under evolving circumstances.

## Author contributions

**Conceptualization:** Sileshi Sintayehu Sharbayta, Youngji Jo, Bruno Buonomo.

**Formal analysis:** Sileshi Sintayehu Sharbayta.

**Funding acquisition:** Youngji Jo, Jaehun Jung.

**Investigation:** Sileshi Sintayehu Sharbayta, Youngji Jo.

**Methodology:** Sileshi Sintayehu Sharbayta, Youngji Jo.

**Project administration:** Youngji Jo, Jaehun Jung.

**Resources:** Jaehun Jung.

**Software:** Sileshi Sintayehu Sharbayta.

**Supervision:** Youngji Jo, Jaehun Jung.

**Validation:** Sileshi Sintayehu Sharbayta.

**Visualization:** Sileshi Sintayehu Sharbayta.

**Writing – original draft:** Sileshi Sintayehu Sharbayta, Youngji Jo.

**Writing – review & editing:** Youngji Jo, Jaehun Jung, Bruno Buonomo.

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
