## [Decision Letter · Decision Letter 0]

31 Mar 2025

Dear Dr. Jo,

Thank you for submitting your manuscript to PLOS ONE. After careful consideration, we feel that it has merit but does not fully meet PLOS ONE’s publication criteria as it currently stands. Therefore, we invite you to submit a revised version of the manuscript that addresses the points raised during the review process.

We look forward to receiving your revised manuscript.

Kind regards,

Eunha Shim

Academic Editor

PLOS ONE

Journal Requirements:

“This research was funded by the Infectious Disease Medical Safety Project, supported by the Ministry of Health & Welfare, Republic of Korea (Grant Number: HG22C0094)”

Reviewers' comments:

Reviewer's Responses to Questions

**Comments to the Author**

1. Is the manuscript technically sound, and do the data support the conclusions?

Reviewer #1: Partly

Reviewer #2: Yes

2. Has the statistical analysis been performed appropriately and rigorously?

Reviewer #1: Yes

Reviewer #2: Yes

3. Have the authors made all data underlying the findings in their manuscript fully available?

Reviewer #1: Yes

Reviewer #2: Yes

4. Is the manuscript presented in an intelligible fashion and written in standard English?

Reviewer #1: Yes

Reviewer #2: Yes

Reviewer #1: The manuscript discusses a novel and highly interesting model for analysis of the relevance of information on the progress of COVID-19. The model formally analyzed and fit to the Omicron wave in South Korea. Numerical scenarios and sensitivity analysis display how the outcomes change when behavioral parameters change.

I am a huge fan of the general concept of the work and the idea of analyzing the impact of human perception on the progress of the disease with a dynamic model. This reveals very interesting partially counterintuitive feedback-loops such as the “more positive tests -> more found cases -> less undetected cases -> less infections -> fewer positive tests”.

Before pointing out my actual scientific concerns, I want to start with a subjective suggestion: in my humble opinion, the authors have overreached themselves a little bit. The introduced model is highly complex and has a huge number of parameters, whereas the desired effects could have been displayed with much less effort. E.g. considering the short time span, effects like natural deaths and births could have been neglected. Also, vaccinations could have been left out without drastically changing the general dynamics of the model. This would have made the model significantly simpler and would have left space for a more rigorous analysis of outcomes other than “active cases”. Less emphasis could also have been placed on model fitting to COVID-19 real numbers without reducing the significance of the paper – on the contrary, the field of application of COIVD-19 could have been extended to a potential future disease X which is currently much more important to the scientific community and the decision makers. However, since this paragraph is only my personal opinion, I leave it to the authors to take it into account or not.

Beside these questions of taste, I have several rather serious general concerns about the used methodology and the significance of the results:

First of all, I need to raise two concerns about the specification of the compartment model, to be precise, with the specification of the IT1, IT2 and H compartments. Concern (a): To me, one of the most interesting features of compartmental models/system dynamics models is, that rates can be interpreted as probabilities and delays (and any mixture of them) at the same time. For parametrization though, it is crucial to distinguish these two. Thus, it is good modelling standard in compartment modelling, that, whenever the flows out of a compartment split, a splitting probability is introduced -> precisely how it is done in E1-> A1/I1. However, this is not done for e.g. A1-> IT1/R. This results in the problem that rho (in your specified model) does not stand for a recovery-rate but rather for the recovery rate multiplied by the chance of not being tested as an asymptomatic - which makes it difficult to interpret. Note that you have the same problem for I1->IT1/R, for IT1 -> R/H and, of course, for the second infection cycle. Concern (b): The property of being detected influences the recovery of an individual which is (to me) unreasonable. Even if the average durations are chosen, so that the recovery time for the tested is, on the average, equivalent with the untested (something like 1/T1 + 1/delta_t = 1/delta), the underlying time distributions differ (the sum of two exponential distributions is not an exponential distribution). Concern (b) is not a big problem for the IT.. -> R/H split, since different recovery times for hospitalized are reasonable (one should think of concern (a) though). However, for the others, something more rigorous needs to be done. The simplest way around would be to add an additional IU1 compartment meaning that all A1 and I1 individuals always flow into IT1 or IU1 first, before they are hospitalized or recover.

The second general concern refers to the message functions. As far as I understood, the choice of the message function is rather arbitrary and somewhat biased. By definition of the function, g1 is always by far larger than g2 meaning that increasing prevalence will always have a much larger impact than increasing hospitalized and deaths. Moreover, deaths enter the functions only via new deaths per day whereas all other quantities enter as active cases which is not fair either. In general, I would argue, that knowledge about (a) an infected (b) a tested infected (c) a hospitalized and (d) a death individual will not only have a different information coverage (e.g. newspapers cannot write about non-tested individuals) but also a different impact w.r. to adherence (increasing death tolls are likely much more influential than increasing case numbers). The authors themselves wrote about this problem and that the model produces counterintuitive results compared to other studies, yet they do not seek for a reason within in their model assumptions.

Some specific comments/concerns:

25) Considering potential publication dates, the first sentence should be updated.

62) The author have properly highlighted several papers in which human adherence/compliance is incorporated. Please explain, how your approach differs from or relates to theirs, respectively.

Figure 1.) Where does the flow for the hospitalized originate in your sketch? It becomes clear after looking at the formulas, but should already be clear when looking at the picture.

132) "<...> regardless of the information." Please specify "the information" in more detail here. I would even suggest to start the section by defining the term "information" and how it is modelled (cal. N and V). This would help understanding how the rates are specified/modelled.

136) "These rates are represented by a constant rate." Sentence sounds strange/unnecessary

163) As far as I understand the formula, the term "average delay <..> to reach the <..> public" is not semantically correct. It rather represents the memory of the general public w.r. to earlier observations - the smaller a, the longer the memory. While this, to some extent, also acts like a delay - the better the public remembers the past, the less important are more recent developments - it should not be confused with a delay: the most recent observations will always influence most (x=0 is the maximum of e^(-ax)). To include an actual delay, a convolution with a Gaussian kernel instead of the exponential would be better suited. [However, I guess, in this case transformation (9) would not be possible though]

178) "where the upper do denotes the time derivative" seems unnecessary considering the target audience of the paper

eq (10)) please add V and N as arguments to F1, T1, F2, T2 to make clear how the equtions are intertwined.

Section 3) In general, some of the introduced parameters in the model specification feel unnecessary, in particular when contrasting them with Table 1. E.g. as far as I understood there is no need to distinguish between B,D,tilde B and tilde D.

242-243) Apparently, birth and death rates have different units. I assume, death rate is per inhabitant, birth rate is in persons per day?! Please state.

245) What does the number 1382042 represent? Does it refer to the maximum theoretical number of available doses per day, the highest ever recorded vaccinations per day or something else? Similar for the tests: considering test-kit availability, is is really realistic that 0.5*55M ~ 22M tests are evaluated per day in South Korea?

Table 1) I find the recovery rate of 1/14 and latency rate of 1/3 hard to believe for Spring 2022 and the prevalent Omicron variant.

Table 1) Please add units!

312) "The result in Figure 3 shows that the model best estimates the observed daily cases when voluntary vaccination and testing are considered." How is this surprising? As far as I understood, the model parameters were fitted to the "responsive" parameters?!

318-322) I don't quite understand, how the effective reproduction number in the fitted model can be smaller than one. If it was, in theory, there should not be a disease wave?! Note: the computed Re should not only depend from the parameters (as stated) but, above all, the initial conditons. Moreover, comparing the computed Re from the model, which is a momentary measure at the start of the simulation, with the "mean reproduction number" from observational studies does not make sense to me.

368) From my intuition is not clear to me, why reactivity to information about prevalence should have a much larger effect than the one for severity. I guess, this originates from the message function issue stated earlier.

Figure 5/6) I understand, that the "active cases", as defined by I1+I2+A1+A2, is important to analyze disease containment, since those are the ones most influential for the spread. However, the tranmissibility of the asymptomatic is smaller than the one for the infected whereas (1-delta) of the detected infected individuals contribute to spreading as well (and with a higher test adherence, this part would increase). So the chosen outcome is in my optinion not optimal to evaluate the "success" of an intervention. Why not take the peak of the hospitalized compartment or the cumulative deaths instead? From the public health aspect this would be more influential anyway.

478) "even the" -> "even IF the"

578) Additional delay time of vaccinations i.e. time between the shot is issued and the vaccination works properly (~14days for typical CoV MRNA vaccines) should be added to the limitation list.

Reviewer #2: This manuscript presents a timely and technically interesting behavioral transmission model of COVID-19, calibrated to data from South Korea during the Omicron wave (February–May 2022). The study makes a valuable contribution by incorporating immunity-based behavioral responses into a compartmental modeling framework and assessing their impact on epidemic dynamics. The use of real-world data for model calibration and the exploration of behavioral heterogeneity adds to its relevance for both public health policy and epidemic modeling literature.

.

Reviewer #1: No

Reviewer #2: **Yes:** Nawa A AlshammariNawa A AlshammariNawa A AlshammariNawa A Alshammari

---

## [Author Response · Author response to Decision Letter 1]

25 Apr 2025

Below is our point-by-point response to each of the reviewers' comments. Corresponding revisions in the manuscript are indicated in each response using line numbers or equation numbers. In the manuscript, all textual changes are highlighted in red.

Reviewer 1:

Subjective suggestions :

The introduced model is highly complex and has a huge number of parameters, whereas the desired effects could have been displayed with much less effort.

For example.

considering the short time span, effects like natural deaths and births could have been neglected.

vaccinations could have been left out without drastically changing the general dynamics of the model.

This would have made the model significantly simpler and would have left space for a more rigorous analysis of outcomes other than “active cases”.

Response: We agree that births and natural deaths are often excluded from short‑term epidemic models; however, adding them here introduces only a few extra parameters, while most complexity arises from our immunity‑ and decision‑based compartments. Thus, we keep them for structural consistency and to facilitate future extensions to longer horizons where demographics may become important.

Regarding the reviewer's comment on vaccination inclusion in our model, we agree that models should indeed remain as simple as possible. However, vaccination behavior is a primary driver of the dynamics we seek to explain, and omitting it would remove a core feedback loop. Our framework distinguishes partially immunized individuals from susceptible individuals, with immunity upgraded by primary vaccination, booster doses, or infection-induced recovery. Because vaccinated individuals can still become infected, their risk perception—and thus testing and revaccination choices—evolves: Individuals who received their primary series some time ago respond more actively to seek care based on current prevalence (rate F1), whereas booster recipients are driven more by perceived disease severity (rate F2). These dynamics create essential interactions between immunity, behavior, and transmission that would be lost if vaccination were excluded from the model.

Less emphasis could also have been placed on model fitting to COVID-19 real numbers without reducing the significance of the paper – on the contrary, the field of application of COIVD-19 could have been extended to a potential future disease X which is currently much more important to the scientific community and the decision makers. However, since this paragraph is only my personal opinion, I leave it to the authors to take it into account or not.

Response: We appreciate the reviewer’s thoughtful suggestion regarding the broader applicability of our model. Although we used COVID 19 as a case study due to the availability of real-world data for model calibration and validation, the model’s core feature is its behavioral feedback tied to evolving immune status, from a susceptible population early in an outbreak to one with widespread vaccination. Capturing how behaviour shifts as immunity accumulates is directly applicable to future pandemics with any emerging pathogen, especially when vaccines or other countermeasures become available mid course. We appreciate this perspective and will consider emphasizing the model’s adaptability in future work.

General concerns

Two concerns about the specification of the compartment model, to be precise, with the specification of the IT1, IT2 and H compartments

To me, one of the most interesting features of compartmental models/system dynamics models is, that rates can be interpreted as probabilities and delays (and any mixture of them) at the same time. For parametrization though, it is crucial to distinguish these two. Thus, it is good modelling standard in compartment modelling, that, whenever the flows out of a compartment split, a splitting probability is introduced -> precisely how it is done in E1-> A1/I1. However, this is not done for e.g. A1-> IT1/R. This results in the problem that rho (in your specified model) does not stand for a recovery-rate but rather for the recovery rate multiplied by the chance of not being tested as an asymptomatic - which makes it difficult to interpret. Note that you have the same problem for I1->IT1/R, for IT1 -> R/H and, of course, for the second infection cycle.

Response: Thank you for highlighting this important modeling issue, particularly its implications for interpreting recovery rates. In response to your insightful comment, and as further elaborated in your point (b), we have revised the model structure to improve clarity and consistency in representing flows out of the infectious compartments.

Specifically, we introduced an additional compartments, IU1 (in primary dynamics) and IU2 (in secondary dynamics), to explicitly represent undetected asymptomatic and symptomatic individuals. With this modification, individuals in compartments A1 (asymptomatic) and I1 (symptomatic) now move to either IT1 (tested) or the new IU1 (untested) compartment, with the split governed by clear probabilities (see revised Fig. 1, p 6, and explained in lines 77 -80.). We assume that individuals in A1 remain in that compartment for a longer average duration (parameterized by a1) compared to I1, before testing occurs. However, both durations are assumed to be short enough that neither recovery nor death occurs within these compartments. Moreover, we assume that individuals in compartments A1 and I1 do not contribute to transmission, as these compartments represent a pre-testing stage with a short duration of stay, during which the likelihood of onward transmission is assumed to be negligible (lines 114 - 120). We have applied the same modeling modification to the second dynamics.

Regarding the transition from IT1 (IT2) to R and H, we have retained the original structure, as we consider recovery and hospitalization to be governed by distinct biological processes, each with its own independent rate. This approach, which avoids the use of a splitting probability in this specific case, is supported by several previous studies, for example:

From identified infected compartment to Hospitalized and Quarantined compartment without splitting probability [1]

From carriers(C) to ill (I) and recovered (R) compartment[2]

From Infected (I) to Hospital (H) and Recovery (R) [3]

We believe these changes clarify parameter interpretation and align the model with standard compartmental conventions while preserving its biological realism.

The property of being detected influences the recovery of an individual which is (to me) unreasonable. Even if the average durations are chosen, so that the recovery time for the tested is, on the average, equivalent with the untested (something like 1/T1 + 1/delta_t = 1/delta), the underlying time distributions differ (the sum of two exponential distributions is not an exponential distribution). Concern (b) is not a big problem for the IT. -> R/H split, since different recovery times for hospitalized are reasonable (one should think of concern (a) though). However, for the others, something more rigorous needs to be done. The simplest way around would be to add an additional IU1 compartment meaning that all A1 and I1 individuals always flow into IT1 or IU1 first, before they are hospitalized or recover.

Response: Thank you for raising this point. We agree that if detection status is allowed to alter recovery dynamics, inconsistencies can arise in the underlying dwell time distributions. Our revision—introducing the IU1 compartment (see response a)—resolves this by routing every individual in A1 or I1 first to IT1 (tested) or IU1 (untested) before any recovery or hospitalisation step, with the same change applied in the secondary dynamics. This structure allows recovery and hospitalization processes to be governed by consistent and independent rates, regardless of detection status, and avoids the problematic assumption that detection directly alters the recovery time distribution. We greatly appreciate your insightful recommendation, which significantly improved the conceptual rigor of the model.

As far as I understood, the choice of the message function is rather arbitrary and somewhat biased. By definition of the function, g1 is always by far larger than g2 meaning that increasing prevalence will always have a much larger impact than increasing hospitalized and deaths. Moreover, deaths enter the functions only via new deaths per day whereas all other quantities enter as active cases which is not fair either. In general, I would argue, that knowledge about (a) an infected (b) a tested infected (c) a hospitalized and (d) a death individual will not only have a different information coverage (e.g. newspapers cannot write about non-tested individuals) but also a different impact w.r. to adherence (increasing death tolls are likely much more influential than increasing case numbers). The authors themselves wrote about this problem and that the model produces counterintuitive results compared to other studies, yet they do not seek for a reason within in their model assumptions.

Response: Thank you for highlighting the imbalance between our two information “message” functions g1 (prevalence) and g2 (severity). Because g1 formerly used both detected and undetected symptomatic cases, it was systematically larger than g2, which relied only on deaths and hospitalizations; this could overstate the behavioral weight of prevalence relative to severity.

To correct this, we made the following two changes in the revised manuscript

We have redefined g1 to depend solely on publicly visible indicators-- tested cases and hospitalizations (Equation 4)

We assumed the reactivity to severity is twofold of the reactivity to prevalence- such assumption reflects the notion that severity-related events are generally more visible to the public and psychologically impactful, often prompting stronger behavioral responses ( lines 306 – 316)

These modification address somehow the dominance of g1 over g2. In general, the limitation that message functions formulation and fixing baseline values to reactivity parameter are based on only theoretical basis is acknowledged (lines 619 – 629)

A sensitivity check varying the severity‑reactivity parameter across a wider range shows the revised formulation reduces g1’s dominance without materially altering peak‑infection projections, thereby improving conceptual balance while preserving overall model behavior.

Regarding the discrepancy between our model and other studies, we acknowledge that varying definitions of the message functions in our model can shift the dominant behavioral drivers and lead to different outcomes as compared to other studies. To make this clear, we have added a brief discussion (lines 595 -599) to provide further context and transparency regarding the assumptions behind our model’s structure could be one the sources of the discrepancy.

Specific comments:

25) Considering potential publication dates, the first sentence should be updated.

Response: Thank you for the suggestion. We have updated the reference and replaced it with a more recent publication to reflect the most current literature and ensure the relevance of the citation (line 4).

62) The author have properly highlighted several papers in which human adherence/compliance is incorporated. Please explain, how your approach differs from or relates to theirs, respectively.

Response: Thank you for prompting a clearer comparison. Like earlier information‑index models, our framework links behaviour to epidemic indicators, but prior work typically treats the population as homogeneous and ties compliance to a single signal—most often prevalence. Our model extends this paradigm in two ways: (i) it distinguishes two information streams—prevalence (g1) and severity (g2)—and allows separate reactivity parameters, and (ii) it embeds heterogeneity by immune status, so vaccinated/boosted and non‑immune individuals can weigh those signals differently. These additions generate richer, more realistic behavioural dynamics while remaining within the information‑index tradition. The revised manuscript now outlines this distinction in lines 40‑56.

Figure 1.) Where does the flow for the hospitalized originate in your sketch? It becomes clear after looking at the formulas, but should already be clear when looking at the picture.

Response: Thank you for noting this. In the diagram (Figure 1), arrows from both tested compartments—IT1 in the primary cycle and IT2 in the secondary cycle—feed into the Hospitalised (H) compartment. We have re‑checked the figure and confirmed that these pathways are indeed present and correctly labelled.

132) "<...> regardless of the information." Please specify "the information" in more detail here. I would even suggest to start the section by defining the term "information" and how it is modelled (cal. N and V). This would help understanding how the rates are specified/modelled.

Response: Thank you for the suggestion. We have added a new subsection, “Information,” immediately before the vaccination and testing rate definitions (lines 142 - 170) to explicitly define the information signals that govern behavioural rates in the model.

136) "These rates are represented by a constant rate." Sentence sounds strange/unnecessary

Response: Thank you for pointing this out. We agree that the original sentence was unclear and potentially unnecessary. Our intention was to clarify that the mandatory components of vaccination and testing rates are modeled as constants, while the voluntary components are dynamics. In the revised manuscript, we now made it clear by only adding a phrase ‘modeled as constant’ as in the sentence ‘First, the mandatory vaccination and testing rates (modeled as constant) represent the rates for the portion of the population that will be vaccinated or tested regardless of the information.’ (line 173)

163) As far as I understand the formula, the term "average delay <..> to reach the <..> public" is not semantically correct. It rather represents the memory of the general public w.r. to earlier observations - the smaller a, the longer the memory. While this, to some extent, also acts like a delay - the better the public remembers the past, the less important are more recent developments - it should not be confused with a delay: the most recent observations will always influence most (x=0 is the maximum of e^(-ax)). To include an actual delay, a convolution with a Gaussian kernel instead of the exponential would be better suited. [However, I guess, in this case transformation (9) would not be possible though]

Response: Thank you for the helpful comment. We now describe parameter a as a memory‑decay coefficient: smaller values mean the public retains past information longer, while larger values indicate faster forgetting. This usage follows earlier information‑index studies [4]. The revised wording appears in the Information Indices section (line 150).

178) "where the upper dot denotes the time derivative" seems unnecessary considering the target audience of the paper

Response: Removed in the revised manuscript

eq (10)) please add V and N as arguments to F1, T1, F2, T2 to make clear how the equations are intertwined.

Response: Thank you for the suggestion. We have updated Eq(10) accordingly in the revised manuscript.

Section 3) In general, some of the introduced parameters in the model specification feel unnecessary, in particular when contrasting them with Table 1. E.g. as far as I understood there is no need to distinguish between B,D,tilde B and tilde D.

Response: Thank you for the comment. Although the model originally introduced four reactivity parameters (B, D, (B,) *~* and D and D and D and D *~* ), we lacked data to estimate them separately, so we treated them as identical. In the revision, we have collapsed them into just two parameters—reactivity to prevalence (D) and to severity (B)—as shown in Eqs. 6–9.), we lacked data to estimate them separately, so we treated them as identical. In the revision, we have collapsed them into just two parameters—reactivity to prevalence (D) and to severity (B)—as shown in Eqs. 6–9.), we lacked data to estimate them separately, so we treated them as identical. In the revision, we have collapsed them into just two parameters—reactivity to prevalence (D) and to severity (B)—as shown in Eqs. 6–9.), we lacked data to estimate them separately, so we treated them as identical. In the revision, we have collapsed them into just two parameters—reactivity to prevalence (D) and to severity (B)—as shown in Eqs. 6–9.

242-243) Apparently, birth and death rates have different units. I assume, death rate is per inhabitant, birth rate is in persons per day?! Please state.

Response: Thank you for pointing this out. We have updated the units for clarity in the revised manuscript. Specific

---

## [Decision Letter · Decision Letter 1]

18 Nov 2025

Dear Dr. Jo,

Thank you for submitting your manuscript to PLOS ONE. After careful consideration, we feel that it has merit but does not fully meet PLOS ONE’s publication criteria as it currently stands. Therefore, we invite you to submit a revised version of the manuscript that addresses the points raised during the review process.

We look forward to receiving your revised manuscript.

Kind regards,

Eunha Shim

Academic Editor

PLOS ONE

Journal Requirements:

Reviewers' comments:

Reviewer's Responses to Questions

**Comments to the Author**

Reviewer #3: (No Response)

Reviewer #4: All comments have been addressed

Reviewer #5: All comments have been addressed

Reviewer #6: (No Response)

2. Is the manuscript technically sound, and do the data support the conclusions?

Reviewer #3: Partly

Reviewer #4: No

Reviewer #5: Yes

Reviewer #6: Yes

3. Has the statistical analysis been performed appropriately and rigorously?

Reviewer #3: Yes

Reviewer #4: Yes

Reviewer #5: Yes

Reviewer #6: Yes

4. Have the authors made all data underlying the findings in their manuscript fully available?

Reviewer #3: Yes

Reviewer #4: Yes

Reviewer #5: Yes

Reviewer #6: Yes

5. Is the manuscript presented in an intelligible fashion and written in standard English?

Reviewer #3: Yes

Reviewer #4: No

Reviewer #5: Yes

Reviewer #6: Yes

Reviewer #3: The presented work is interesting and unifies several ideas of extensions of SIR type ODE models to a 15 compartment SEIR type model that partitions the population into a (naive) primary and a (partially immune) secondary branch. Vaccination and testing rates each comprise a constant mandatory component and a voluntary component governed by two information indices for prevalence and severity. The parameters are estimated by data of the Republic of Korea from 1 Feb to 31 May 2022 and explore how information related parameters influence epidemic peaks. In contrast to agent-based models of the whole population this approach allows more analytic analyses which are presented nicely. However, there is also the drawback of less flexibility (preventing state explosion) and the formulas lack the co-existence of several strains, which for example was especially relevant when the Omicron BA.1 and BA.2 strain emerged (which falls into the period of the collected data), leading do very different immune response behaviour (both after infection and vaccination) for the affected population groups. The quality of the paper could be increased if these and other limitations, and what they mean for policy makers, would be discussed in more detail.

Please also consider clarifying the following points:

o l. 262–269: Please provide seroprevalence or administrative vaccination coverage data to substantiate the 85 % figure on 1 Feb 2022. Sensitivity analysis in Table 1 varies biological parameters but not this initial condition, but Figure 3 shows the model dynamics are highly sensitive to the susceptible fraction. This uncertainty is even more prevalent as the calibration period falls directly in the early (and in some countries double-) Omicron-waves.

o Only four parameters are fitted using vaccination and testing data. Because mandatory and voluntary rates are both fitted indirectly, their effects are confounded. Please provide confidence intervals or further distributions to validate the fitted parameters.

o In equation (13) voluntary testing does not appear, even though the force of infection depends on the isolated fraction, therefore the following results may not be true, so either include this fraction or explain why the simplification is legitimate.

o Figure 2b shows the fitted model results, however it is hard to assess the goodness of fit without metrics, so provide RMSE or another quantitative fit. Also, include observed vs. fitted daily testing numbers.

o Some statements need more explanation or clarification when they are true, for example “increasing reactivity from 50 % to 100 % would reduce the peak by 16 %” is only true under the assumption of fixed mandatory rates. Also, reactivity is not an intervention, but a behavioural parameter, thus confidence intervals (or other measures of uncertainty) are recommended.

o There are either some typos, the paper should be proof-read once more.

o Provide a complete table of symbols.

o The assumption from [34] that the number of exposed people is 20x the number of infected people seems wrong. The source [34] is not the initial source, [34] references to sources from early 2020. Other simulation studies use much lower ratios for exposed to infected. Please verify that or explain that assumption.

The manuscript is promising but requires some methodological clarification and stronger justification of assumptions. Also, it needs clarification under which prerequisites and in which phase of the pandemic their results are applicable for policy makers, including a discussion of the limitations, especially after entering the endemic phase.

Reviewer #4: Title: The Impact of Human Behavioral Adaptation Stratified by Immune Status on COVID-19 Spread,

with Application to South Korea

This study develops a behavioral transmission model to examine how COVID-19 spread differs

between partially immune and susceptible individuals. Using data from South Korea (February–May

2022), the model incorporates variations in vaccination response, testing behavior, and reliance on

information about disease prevalence and severity. Simulations show that if partially immune

individuals react to information as strongly as susceptible individuals, peak active cases may

decrease by 16%. However, shifting their risk perception from prevalence to severity could increase

the peak by 50%. These results emphasize the importance of adaptive vaccination and testing

strategies as public risk perceptions change in the post-pandemic era. The paper has the following

issues:

• Table 1 contains several errors.

• Figure 4 appears to be incorrect.

• Figure 6 is not accurate and requires correction.

• The overall English quality of the manuscript is very weak and needs thorough revision.

• The references are outdated; please update them with recent works from 2023–2025.

• The following references should be consulted and incorporated into the study:

• DOI: 10.3934/math.2025986 ; DOI: 10.3390/mca30050100;

DOI: 10.1371/journal.pone.0331243; DOI: 10.3390/math13172822

• What is the role of *θ* in Figure 6? Please clarify its meaning and contribution to the analysis.in Figure 6? Please clarify its meaning and contribution to the analysis.in Figure 6? Please clarify its meaning and contribution to the analysis.in Figure 6? Please clarify its meaning and contribution to the analysis.

• Explain the significance of the parameters F10, F20, T10, and T2 used in the figures. What do

they represent, and how do they influence the model?

Reviewer #5: (No Response)

Reviewer #6: The paper describes a thorough epidemiological analysis of Covid-19 spread in South Korea, using multi-compartment models that incorporate the kinetics of transitioning from one compartment to another. Although I agree with reviewer 1 that the model is quite complex, I think the methods are quite detailed and could be informative. It looks like the comments from the previous review cycle were addressed adequately. The only other comment I had was for Table 1. All the parameter values indicated do not have any standard error associated with them. It could be worth addressing how deviations in the parameter value assumptions for those parameters, to which the model is especially sensitive to, would affect the results and/or its interpretation.

.

Reviewer #3: No

Reviewer #4: No

Reviewer #5: No

Reviewer #6: No

---

## [Author Response · Author response to Decision Letter 2]

28 Dec 2025

The manuscript has been revised to address the reviewers’ major comments by strengthening methodological clarity, transparency, and presentation. Key revisions include: (i) clearer discussion of model scope, assumptions, and limitations, particularly regarding behavioral responses, applicability across epidemic versus endemic phases, and the exclusion of multiple co-circulating variants; (ii) improved parameter estimation and uncertainty quantification through the addition of 95% confidence intervals for fitted parameters and a dedicated sensitivity analysis subsection describing the methods used; (iii) inclusion of a new Scenario subsection in the Methods to clearly define and organize the simulated behavioral and policy scenarios; and (iv) enhanced model validation and presentation, including quantitative goodness-of-fit metrics, clarification of daily testing dynamics, correction and reformatting of figures and tables, and the addition of a complete table of symbols and parameters. References were updated and the manuscript was thoroughly proofread to improve clarity and consistency.

Below, we provide a point-by-point response to each reviewer comment. All new changes introduced in the present revision are highlighted in blue, while modifications made during the first revision are retained and shown in red in the manuscript.

Reviewer-3

The presented work is interesting and unifies several ideas of extensions of SIR type ODE models to a 15 compartment SEIR type model that partitions the population into a (naive) primary and a (partially immune) secondary branch. Vaccination and testing rates each comprise a constant mandatory component and a voluntary component governed by two information indices for prevalence and severity. The parameters are estimated by data of the Republic of Korea from 1 Feb to 31 May 2022 and explore how information related parameters influence epidemic peaks. In contrast to agent-based models of the whole population this approach allows more analytic analyses which are presented nicely. However, there is also the drawback of less flexibility (preventing state explosion) and the formulas lack the co-existence of several strains, which for example was especially relevant when the Omicron BA.1 and BA.2 strain emerged (which falls into the period of the collected data), leading do very different immune response behaviour (both after infection and vaccination) for the affected population groups. The quality of the paper could be increased if these and other limitations, and what they mean for policy makers, would be discussed in more detail.

Reply: We sincerely thank the reviewer for the insightful and well-considered feedback. We agree that incorporating co-circulating variants (such as Omicron BA.1 and BA.2) and additional immunity states can enhance realism, but doing so also substantially increases the number of compartments and parameters (e.g., strain-specific infection, recovery, and immune evasion rates). Given the primary focus of this study was to isolate the impact of behavioral adaptation based on immune status (i.e. How information-driven voluntary vaccination and testing behaviors shape epidemic dynamics through differential responsiveness among immunologically naïve and partially immune individuals), we opted for a parsimonious structure to maintain mathematical tractability and avoid the uncertainty associated with parameterizing multiple co-circulating strains without granular strain-specific behavioral data.

262–269: Please provide seroprevalence or administrative vaccination coverage data to substantiate the 85 % figure on 1 Feb 2022. Sensitivity analysis in Table 1 varies biological parameters but not this initial condition, but Figure 3 shows the model dynamics are highly sensitive to the susceptible fraction. This uncertainty is even more prevalent as the calibration period falls directly in the early (and in some countries double-) Omicron-waves.

Reply: First of all we would like to clarify that Table 1 reports baseline parameter values and does not represent a sensitivity analysis. Figure 3 illustrates the model fit to observed daily cases rather than parameter sensitivity. We have added a reference supporting the assumption of an 85 % susceptible fraction on 1 Feb 2022, based on seroprevalence study in South Korea (Line 313-324). However, we acknowledge the uncertainty of vaccination coverage of 85% (fraction of initial partially immune population (S2+S3) ranging from 70% to 98%) with the waning immunity), and added this into sensitivity analysis (Fig 8) and corresponding explanation in the method section (lines 427-429).

Only four parameters are fitted using vaccination and testing data. Because mandatory and voluntary rates are both fitted indirectly, their effects are confounded. Please provide confidence intervals or further distributions to validate the fitted parameters.

Reply: Thank you for this constructive comment. Although mandatory and voluntary rates jointly shape vaccination and testing dynamics, they enter the model as additive processes that cannot be independently identified from aggregate data. Our fitting process therefore focuses on estimating the combined effect captured by the four fitted parameters (e.g. *β*_1,_1,_1,_1,*β*_2,k,_2,k,_2,k,_2,k,*ξ*). In the revised manuscript, we now report 95% confidence intervals for all fitted parameters (See Table 2). These intervals were calculated using the covariance matrix produced by the nonlinear least-squares fitting algorithm (curve_fit), which provides an estimate of the uncertainty surrounding each parameter. A brief explanation of the procedure has been included in the Model fitting and parameter estimation section (Lines 365–368).). In the revised manuscript, we now report 95% confidence intervals for all fitted parameters (See Table 2). These intervals were calculated using the covariance matrix produced by the nonlinear least-squares fitting algorithm (curve_fit), which provides an estimate of the uncertainty surrounding each parameter. A brief explanation of the procedure has been included in the Model fitting and parameter estimation section (Lines 365–368).). In the revised manuscript, we now report 95% confidence intervals for all fitted parameters (See Table 2). These intervals were calculated using the covariance matrix produced by the nonlinear least-squares fitting algorithm (curve_fit), which provides an estimate of the uncertainty surrounding each parameter. A brief explanation of the procedure has been included in the Model fitting and parameter estimation section (Lines 365–368).). In the revised manuscript, we now report 95% confidence intervals for all fitted parameters (See Table 2). These intervals were calculated using the covariance matrix produced by the nonlinear least-squares fitting algorithm (curve_fit), which provides an estimate of the uncertainty surrounding each parameter. A brief explanation of the procedure has been included in the Model fitting and parameter estimation section (Lines 365–368).

In equation (13) voluntary testing does not appear, even though the force of infection depends on the isolated fraction, therefore the following results may not be true, so either include this fraction or explain why the simplification is legitimate.

Reply: We appreciate the reviewer’s comment. Because the next-generation matrix is computed at the disease-free equilibrium, the information variables of voluntary vaccination/testing terms evaluate to zero ( V=N=0), in prevalence-dependent functions and the testing/vaccination rates reduce to their baseline constants (T10, T20 or F10, F20) in estimating the effective reproduction number R_e. We have added a detailed explanation to the revised manuscript (Lines 227–236).

Figure 2b shows the fitted model results, however it is hard to assess the goodness of fit without metrics, so provide RMSE or another quantitative fit. Also, include observed vs. fitted daily testing numbers.

Reply: We thank the reviewer for this constructive suggestion. We would like to clarify that the reviewer is referring to the version of the manuscript submitted during the first review cycle, in which the fitting results appeared as Figure 2. In the revised manuscript (submitted after the first round of review), substantial restructuring was done, and the corresponding results now appear in Figure 3. In the revised manuscript, we now explicitly quantify the goodness of fit by reporting normalized RMSE values (based on min–max scaling) for each data stream. The resulting normalized RMSE values (0.022 for cumulative cases, 0.091 for vaccinations, and 0.015 for deaths.) indicate that the model attains comparable relative accuracy for cumulative cases and deaths, while the fit to cumulative vaccination data is somewhat less precise (Lines 368–375). Nonetheless, the overall fitting performance remains strong, and the model successfully reproduces the major temporal patterns across all three data sets despite differences in scale.

Regarding daily testing, we note that the cumulative case fitting is inherently based on daily detected cases (I_T1+I_T2). As presented in the manuscript, Figure 4 shows the observed daily tested cases along with the corresponding model approximation. This figure illustrates the temporal pattern of model-predicted daily testing and demonstrates how the model reproduces fluctuations in detected cases over time, complementing the cumulative fitting shown in Figure 3a. In the revised manuscript, we have clarified this point to clearly indicate how Figure 4 complements the cumulative case fitting presented in Figure 3a (Lines 390-397).

Some statements need more explanation or clarification when they are true, for example “increasing reactivity from 50 % to 100 % would reduce the peak by 16 %” is only true under the assumption of fixed mandatory rates. Also, reactivity is not an intervention, but a behavioural parameter, thus confidence intervals (or other measures of uncertainty) are recommended.

Reply: We thank the reviewer for this helpful comment. It appears that the reviewer is referring to an earlier version of the manuscript, in which the illustrative example reported a reduction of 16% when increasing the reactivity of partially immune individuals from 50% to 100%. In the revised manuscript, this result has been updated based on the corrected simulations and now states that such an increase leads to a 15% reduction relative to the baseline, and a 33% reduction compared to the scenario in which partially immune individuals do not react to information at all (*θ* = 0). We agree that the effects of the voluntary components are observed under the assumption of fixed mandatory rates. To clarify this, we have revised the discussion section to explicitly state the conditions under which these effects occur. Additionally, we note that uncertainty in this behavioral parameter is explored through parameter variation and further assessed via one-way sensitivity analysis (Lines 583–593).= 0). We agree that the effects of the voluntary components are observed under the assumption of fixed mandatory rates. To clarify this, we have revised the discussion section to explicitly state the conditions under which these effects occur. Additionally, we note that uncertainty in this behavioral parameter is explored through parameter variation and further assessed via one-way sensitivity analysis (Lines 583–593).= 0). We agree that the effects of the voluntary components are observed under the assumption of fixed mandatory rates. To clarify this, we have revised the discussion section to explicitly state the conditions under which these effects occur. Additionally, we note that uncertainty in this behavioral parameter is explored through parameter variation and further assessed via one-way sensitivity analysis (Lines 583–593).= 0). We agree that the effects of the voluntary components are observed under the assumption of fixed mandatory rates. To clarify this, we have revised the discussion section to explicitly state the conditions under which these effects occur. Additionally, we note that uncertainty in this behavioral parameter is explored through parameter variation and further assessed via one-way sensitivity analysis (Lines 583–593).

There are either some typos, the paper should be proof-read once more.

Reply: We thank the reviewer for this observation. The manuscript has been carefully proofread and revised to correct typos and improve overall clarity.

Provide a complete table of symbols.

Reply: We appreciate the reviewer’s comment. In the revised manuscript, we have now added a complete and consolidated table listing all compartment symbols and parameters along with their descriptions (Table 1) and we provided the parameter’s baseline values in a separate Table (Table 2). Table 2 also reports parameter ranges, including 95% confidence intervals obtained from the literature when available and from the model-fitting procedure, as well as ranges defined as ±25% of the baseline values used in the sensitivity analysis. We believe this addition improves clarity and makes the model structure easier to follow.

The assumption from [34] that the number of exposed people is 20x the number of infected people seems wrong. The source [34] is not the initial source, [34] references to sources from early 2020. Other simulation studies use much lower ratios for exposed to infected. Please verify that or explain that assumption.

Reply: We thank the reviewer for pointing this out. We acknowledge that this ratio could be higher than in some other modeling studies, which often use lower exposed-to-infected ratios or treat the initial exposed population as a free parameter. In our case, we intentionally selected a larger initial exposed population to account for the possibility of substantial undetected or presymptomatic infections at the start of the simulation. Although this choice leads to an overestimation of the initial exposed compartment, the model is initialized with zero undetected infectious individuals; therefore, the large exposed count naturally transitions into undetected infections during the early phase of the simulation. We have clarified this rationale in the revised manuscript (Lines 329–337).

The manuscript is promising but requires some methodological clarification and stronger justification of assumptions. Also, it needs clarification under which prerequisites and in which phase of the pandemic their results are applicable for policy makers, including a discussion of the limitations, especially after entering the endemic phase.

Reply: We thank the reviewer for this insightful comment. In the revised manuscript, we have clarified the specific context and prerequisites under which our results are applicable namely, epidemic-wave periods characterized by rapid transmission, active testing, and dynamic behavioral responses. We have also added a brief explanation regarding the more limited quantitative applicability of our findings during the stable endemic phase. These points are now explicitly noted in both the Discussion and the Limitations sections to better guide interpretation of our results (Lines 557-563, 605-611, 702-706).

Reviewer 4

This study develops a behavioral transmission model to examine how COVID-19 spread differs between partially immune and susceptible individuals. Using data from South Korea (February–May 2022), the model incorporates variations in vaccination response, testing behavior, and reliance on information about disease prevalence and severity. Simulations show that if partially immune individuals react to information as strongly as susceptible individuals, peak active cases may

decrease by 16%. However, shifting their risk perception from prevalence to severity could increase the peak by 50%. These results emphasize the importance of adaptive vaccination and testing strategies as public risk perceptions change in the post-pandemic era. The paper has the following issues:

Reply: We would like to clarify that it appears that the reviewer is referring to an earlier version of the manuscript, in which we reported a reduction of 16% when increasing the reactivity of partially immune individuals from 50% to 100%. In the revised manuscript (submitted during the first-round revision), this result has been updated based on the corrected simulations and now states that such an increase leads to a 15% reduction relative to the baseline, and a 33% reduction compared to the scenario in which partially immune individuals do not react to information at all (*θ* = 0). Moreover, shift in the risk perception from prevalence to severity increases the peak by 57%.= 0). Moreover, shift in the risk perception from prevalence to severity increases the peak by 57%.= 0). Moreover, shift in the risk perception from prevalence to severity increases the peak by 57%.= 0). Moreover, shift in the risk perception from prevalence to severity increases the peak by 57%.

Table 1 contains several errors

Reply: While the specific errors referred to were not clearly identified, we carefully revised Table 1 and the associated parameter list. In doing so, we identified a few inconsistencies between the parameter (F_10,F_20,T_10,T_20) values reported in the text and those presented in the table. These have now been corrected. In the revised manuscript, we also improved the table formatting for clarity and reorganized the parameter and symbol descriptions into a separate table (Table 1), following a suggestion from another reviewer and parameter values are reported in a separate Table (Table 2). We hope these revisions resolve any confusion the reviewer may have encountered.

Figure 4 appears to be incorrect.

Reply: We thank the reviewer for this comment. However, based on the general comment of the reviewer it appears that the reviewer is referring the original submission prior to the first revision. Therefore, Figure 4 presented the stability region analysis, which was renumbered as Figure 2 during the first revision. Although the specific issue with Figure 4 was not explicitly identified, we carefully re-examined the figure (Figure2 panel a-modified stability figure-during the first revision) and identified a resolution-related issue in the contour plot. Specifically, the initial contour resolution of 20 levels did not fully capture the maximum value of R_eat the baseline (T_10,T_20)=(0.2,0.2)reported in the text (R_e=1.24), resulting in a lower apparent maximum. This has now been corrected by increasing the contour resolution to 100 levels, ensuring that the value of R_eis accurately represented across the parameter space. Additionally, we validated the corrected stability region by showing that trajectories of disease prevalence behave as expected: declining to zero when R_e<1and converging to a non-zero endemic equilibrium when R_e>1 (In the revised version-Figure 2, panel b). This confirms the accuracy of the updated stability region in Figure 2.

Figure 6 is not accurate and requires correction.

Reply: We thank the reviewer for the comment. Although it was not clear to us which specific aspec

---

## [Decision Letter · Decision Letter 2]

10 Mar 2026

The impact of human behavioral adaptation stratified by immune status on COVID-19 spread with application to South Korea

PONE-D-24-53603R2

Dear Dr. Jo,

We’re pleased to inform you that your manuscript has been judged scientifically suitable for publication and will be formally accepted for publication once it meets all outstanding technical requirements.

Kind regards,

Eunha Shim

Academic Editor

PLOS One

Additional Editor Comments (optional):

Reviewers' comments:

Reviewer's Responses to Questions

**Comments to the Author**

Reviewer #4: All comments have been addressed

Reviewer #7: All comments have been addressed

Reviewer #8: All comments have been addressed

2. Is the manuscript technically sound, and do the data support the conclusions?

Reviewer #4: Partly

Reviewer #7: Yes

Reviewer #8: (No Response)

3. Has the statistical analysis been performed appropriately and rigorously?

Reviewer #4: Yes

Reviewer #7: Yes

Reviewer #8: Yes

4. Have the authors made all data underlying the findings in their manuscript fully available?

Reviewer #4: (No Response)

Reviewer #7: Yes

Reviewer #8: Yes

5. Is the manuscript presented in an intelligible fashion and written in standard English?

Reviewer #4: Yes

Reviewer #7: Yes

Reviewer #8: Yes

Reviewer #4: The authors have addressed all the reviewer comments and revised the manuscript accordingly. In its present form, the paper can be accepted for publication.

Reviewer #7: In this study, the authors present a comprehensive statistical analysis of the impact of serological testing and vaccination on the clinical epidemiology of COVID-19. The statistical modeling approach has been applied effectively, enabling the derivation of objective, well-supported conclusions about the role of these parameters in mitigating infection dynamics. The authors have addressed the reviewers’ comments thoroughly and with appropriate scientific rigor. In my assessment, the revised and final version of the manuscript is now suitable for publication.

Reviewer #8: This study develops a behavioral transmission model to examine how COVID-19 spread and care-seeking behavior vary depending on individuals’ immunity status. The population is divided into partially immune and susceptible groups, whose vaccination decisions, testing behavior, and responses to information about disease prevalence and severity differ. Using COVID-19 data from South Korea (Feb 1–May 31, 2022), the model was calibrated to analyze these behavioral dynamics. Results show that if partially immune individuals react to risk information as strongly as susceptible individuals, peak active cases could decrease by about 33%. However, if their risk perception shifts from focusing mainly on prevalence to severity, peak cases could increase by about 57%. The study highlights the importance of adaptive vaccination and testing strategies and shows that differences in risk perception and immunity can significantly influence future infection waves.

The manuscript presents interesting work and provides valuable insights into the COVID-19 pandemic. As the paper has already been reviewed by another reviewer and the authors have addressed most of the comments adequately, I have also examined the responses and found that they satisfactorily clarify potential my concerns as well. Therefore, in my opinion, the manuscript is suitable for publication.

.

Reviewer #4: No

Reviewer #7: No

Reviewer #8: No

---

## [Editor Report · Acceptance letter]

PONE-D-24-53603R2

PLOS One

Dear Dr. Jo,

I'm pleased to inform you that your manuscript has been deemed suitable for publication in PLOS One. Congratulations! Your manuscript is now being handed over to our production team.

Kind regards,

on behalf of

Dr. Eunha Shim

Academic Editor

PLOS One